# Can the detection dog alert on COVID-19 positive persons by sniffing axillary sweat samples? A proof-of-concept study

Dominique Grandjean[1]☯*, Riad Sarkis[2]☯, Clothilde Lecoq-Julien[1]☯, Aymeric Benard[3]☯, Vinciane Roger[1]‡, Eric Levesque[4]‡, Eric Bernes-Luciani[3]‡, Bruno Maestracci[3], Pascal Morvan[5]‡, Eric Gully[6]‡, David Berceau-Falancourt[7]‡, Pierre Haufstater[8], Gregory Herin[9], Joaquin Cabrera[1]‡, Quentin Muzzin[1]‡, Capucine Gallet[1]‡, Hélène Bacqué[1], Jean-Marie Broc[10], Leo Thomas[10], Anthony Lichaa[2], Georges Moujaes[2], Michele Saliba[2], Aurore Kuhn[10], Mathilde Galey[10], Benoit Berthail[10], Lucien Lapeyre[4], Anthoni Capelli[5]‡, Steevens Renault[5]‡, Karim Bachir[5]‡, Anthony Kovinger[5]‡, Eric Comas[6], Aymeric Stainmesse[6]‡, Erwan Etienne[6]‡, Sébastien Voeltzel[6]‡, Sofiane Mansouri[6]‡, Marlène Berceau-Falancourt[11]‡, Aimé Dami[3], Lary Charlet[6], Eric Ruau[6], Mario Issa[2], Carine Grenet[12], Christophe Billy[13], Jean-Pierre Tourtier[4]☯, Loïc Desquilbet[1]☯

1 Ecole Nationale Vétérinaire d'Alfort, Université Paris Est, Maisons-Alfort, France, 2 Université Franco-Libanaise St Joseph, Faculté de Médecine, Beirout, Lebanon, 3 Service d'Incendie et de Secours de Corse du Sud, Ajaccio, France, 4 Centre Hospitalo-Universitaire Henri-Mondor, Université Paris Est-Créteil, Maisons-Alfort, France, 5 DiagNose, Neuvy, France, 6 Service Départemental d'Incendie et de Secours de Seine et Marne, Melun, France, 7 Compagnie Cynophile de la Préfecture de Police, Paris, France, 8 Hôpital Universitaire Pitié-Salpêtrière, Paris, France, 9 Biodesive SAS, Strasbourg, France, 10 Hôpital d'Instruction des Armées Bégin, Saint-Mandé, France, 11 Cynopro Dectection Dogs, Paris, France, 12 Grand Hôpital de l'Est Francilien, Jossigny, France, 13 Centre Hospitalier François Quesnay, Mantes-la-Jolie, France

☯ These authors contributed equally to this work.
‡ These authors also contributed equally to this work.
* dominique.grandjean@vet-alfort.fr

## Abstract

The aim of this proof-of-concept study was to evaluate if trained dogs could discriminate between sweat samples from symptomatic COVID-19 positive individuals (SARS-CoV-2 PCR positive) and those from asymptomatic COVID-19 negative individuals. The study was conducted at 2 sites (Paris, France, and Beirut, Lebanon), followed the same training and testing protocols, and involved six detection dogs (three explosive detection dogs, one search and rescue dog, and two colon cancer detection dogs). A total of 177 individuals were recruited for the study (95 symptomatic COVID-19 positive and 82 asymptomatic COVID-19 negative individuals) from five hospitals, and one underarm sweat sample per individual was collected. The dog training sessions lasted between one and three weeks. Once trained, the dog had to mark the COVID-19 positive sample randomly placed behind one of three or four olfactory cones (the other cones contained at least one COVID-19 negative sample and between zero and two mocks). During the testing session, a COVID-19 positive sample could be used up to a maximum of three times for one dog. The dog and its handler were both blinded to the COVID-positive sample location. The success rate per dog (i.e., the number of correct indications divided by the number of trials) ranged from 76% to 100%. The lower bound of the 95% confidence interval of the estimated success rate was most of the time higher than the success rate obtained by chance after removing the number

**Data Availability Statement:** Data are provided as Supporting information file.

**Funding:** DiagNose, Cynopro Detection Dogs volunteered their dogs to participate to the study

without any financial request. DiagNose, Cynopro Detection Dogs provided support in the form of salaries for authors [PM, MBF, AC, SR, KB], but did not have any additional role in the study design, data collection and analysis, decision to publish, or preparation of the manuscript. Additionally, ICTS Europe, Biodesiv SAS, and Mario K9 also provided support in the form of salaries for authors [PM, GH, MI], but did not have any additional role in the study design, data collection and analysis, decision to publish, or preparation of the manuscript. The specific roles of these authors are articulated in the 'author contributions' section.

**Competing interests:** DiagNose, Cynopro Detection Dogs, ICTS Europe, Biodesiv SAS, and Mario K9 provided support in the form of salaries for authors. This does not alter our adherence to PLOS ONE policies on sharing data and materials. There are no patents, products in development or marketed products associated with this research to declare.

of mocks from calculations. These results provide some evidence that detection dogs may be able to discriminate between sweat samples from symptomatic COVID-19 individuals and those from asymptomatic COVID-19 negative individuals. However, due to the limitations of this proof-of-concept study (including using some COVID-19 samples more than once and potential confounding biases), these results must be confirmed in validation studies.

## Introduction

The "One Health—One Medicine" concept is currently more important than ever as it is bringing medical doctors, veterinary surgeons, epidemiologists, and dog handlers together to share their knowledge and experience in an attempt to combat the SARS-CoV-2 pandemic. Quick, reliable, and widespread testing is one of the key measures to control the pandemic. In June 2020, some European countries managed to control the COVID-19 epidemic by maintaining a low number of COVID-19 positive cases. In this context, the timing of testing is crucial. A high rate of testing is more effective at slowing further outbreaks if conducted early, when the disease prevalence is lower [1].

There is now some evidence that dogs may play a critical role in detecting infectious and parasite diseases as well as cancers [2]. Some studies involving detection dogs suggested that detection of non-infectious diseases by dogs was comparable with standard diagnostic methods [3], especially for cancer detection [4, 5]. In 1989, Williams and Pembroke suggested that dogs may be able to detect malignant tumours based on their specific odour [6]. The first clinical investigation of lung and breast cancers was published by McCulloch et al. in 2006 [7]. In 2010, Willis et al. published a clinical investigation on bladder cancer [8], after having published a proof-of-principle study in 2004 [9]. Subsequent studies have been performed to investigate the ability of dogs to detect colorectal cancer [10], lung [11–13], prostate [14–16], and liver [17] cancers, and melanoma [18, 19]. Regarding other non-infectious diseases, several studies have suggested that dogs can be used as "alert dogs" for diabetic [20–22] and epileptic patients [23], to improve patient quality of life.

Several studies have used dogs to differentiate a range of target insect and parasite odours. For example, Wallner and Ellis [24] trained dogs to locate gypsy moth eggs with a 95% success rate. Richards et al. assessed the dogs' ability to differentiate between nematodes infected and uninfected sheep faeces [25]. Guest et al. showed that trained dogs can use olfaction to identify people with malaria [26]. Bacteriological diseases can also be detected by dogs [27, 28]. Dogs seem able to differentiate cellular cultures infected by the virus causing bovine mucosal disease from non-infected cultures, or from those infected by other viruses (bovine herpes 1 or bovine parainfluenza 3) [29].

As pointed out by Edwards et al. in 2017, "one of the largest threats to the validity of the results from [olfactory detection dog] studies is the possibility that systematic differences, other than disease status, between positive and control samples in both training and testing phases were responsible for the obtained results." [4]. Therefore, the current amount of evidence suggesting that dogs can be used as a complementary tool to detect infectious and non-infectious diseases may not be as high as the number of published studies on the subject suggests. In order to accurately demonstrate that dogs can detect the studied disease, a rigorous protocol must be followed to prevent biases and over-interpretation of the results. Protocol recommendations include ensuring the dog handler is unaware of the individual's disease status when presenting to the dog (i.e., single-blind criterion; the double-blind criterion where all other staff present during the testing procedure are also blinded strengthens the evidence but

may add difficulties when rewarding the dog), ensuring the dog is presented one sample no more than once during the training and the testing sessions (the dog may memorise just the odour, instead of generalising it; this criterion includes the use of different samples between training and testing sessions), ensuring control samples are comparable to positive samples except for disease status (to avoid confounding bias), and randomising sample positions in the line-up (if line-ups are used) [4, 30, 31].

The Nosaïs project, conducted by the UMES (Unité de Médecine de l'Elevage et du Sport) at the National Veterinary School of Alfort (Maisons-Alfort, France), has been established to develop the scientific approach of medical detection dogs. In May 2020, due to the potential occurrence of a second COVID-19 wave in many countries, the Nosaïs team initiated a multicentre proof-of-concept study on the olfactive detection of COVID-19 positive individuals by dogs. This proof-of-concept study is based on the preliminary assumption that dogs could be trained to discriminate between COVID-19 positive and negative individuals due to their strong olfactory acuity. This assumption has been reinforced by the results from a recent pilot study on the olfactory identification of COVID-19 positive individuals by detection dogs [32]. Our assumption was based on the potential excretion of specific catabolites in the sweat, induced by SARS-CoV-2 cellular actions or replications in the organism's cells, through the apocrine sweat glands, generating Volatile Organic Compounds (VOCs) that the dogs can detect.

VOCs are volatile at an ambient temperature, and may be detectable by the dogs, in breath, urine, tears, saliva, faeces and sweat. Most studies on volatile biomarkers have been conducted on breath samples. VOCs emanating from the skin contribute to an individual's body odour, and may convey important information about metabolic processes [33, 34]. They are produced by eccrine, apocrine and sebaceous gland secretions, and are the major source of underarm odorants, playing a role in chemical signalling [35]. Several studies have been performed to better characterise VOCs produced by axillary apocrine glands [36–39]. Sweat from the palms of the hands, soles of the feet and the forehead mainly comes from eccrine glands and sebum. Most of these sweat compounds are organic acids ranging in carbon size from C2 to C20, the most abundant being saturated, monounsaturated and di-unsaturated C16 and C18, which are not volatile at body temperature [40]. More recent studies confirmed that human sweat compounds differ according to the anatomic site [41], and that the secretory capacity of eccrine sweat glands appears larger on the trunk (including underarms) compared to other parts of the body [42].

## Material and methods

This study was carried out in strict accordance with the recommendations in the Guide for the Care and Use of Animals edicted by french law (articles R214-87 to R214-137 of the rural code) updated by decree 2013–118 and 5 decrees edited on february 1st 2013. The protocol was approved by the Committee on the Ethics of Animal Experiments of the Ecole Nationale Veterinaire d'Alfort and by the Protection of Persons Committee of Assistance Publique Hopitaux de Paris (Protocol Covidog of global study Covidef). No invasive study was performed on the dogs participating to the study.

Institutional review board: APHP (Assistance Publique Hopitaux de Paris) Biosafety committee approval: ENVA (Ecole Nationale Veterinaire d'Alfort) Ethic committee approval: ENVA (Ecole Nationale Veterinaire d'Alfort) Scientific committee approval: APHP (Assistance Publique Hopitaux de Paris) and Hopital d'Instruction des Armees Begin Approval of the research and form of consent obtained: name "COVIDOG" by APHP (Assistance Publique Hopitaux de Paris global project "COVIDeF").

### Inclusion criteria of COVID-19 negative and positive individuals

This is a two-site study (National Veterinary School of Alfort, Maisons-Alfort, France, and French-Lebanese University Saint Joseph, College of Medicine, Beirut, Lebanon) which recruited COVID-19 negative and positive individuals from four hospitals in Paris, France (Hôpital d'Instruction des Armées Bégin, Saint-Mandé; Centre Hospitalier François Quesnay, Mantes-la-Jolie; Grand Hôpital de l'Est Francilien, Jossigny; Hôpital Henri-Mondor, Créteil) and one hospital in Beirut, Lebanon (Hôpital Hôtel Dieu de France, Beirut).

Inclusion criteria for COVID-19 positive individuals included showing COVID-19 clinical symptoms and being COVID-19 positive on RT-PCR or PCR test for SARS-CoV-2. To avoid potential interferences with long-term medical treatments in the sweat VOCs, COVID-19 positive individuals were not included in the study if they had received a medical treatment for more than 36 hours prior to the PCR test.

To avoid potential confounding due to "hospital odour" [5, 43], when one COVID-19 positive individual was recruited into the study, a COVID-19 negative individual was recruited from the same hospital who showed no clinical symptoms related to COVID-19 and was COVID-19 negative on the SARS-CoV-2 PCR test. These COVID-19 negative individuals were patients hospitalised for reasons other than COVID-19 symptoms or nurses working in the same hospital.

### Samples

Sweat samples (whether collected from COVID-19 positive or negative individuals) were collected from the underarm region by doctors, interns or nurses, with helpers, who were trained not to contaminate the samples with their own odours. A training video and the necessary equipment were provided to the clinical staff collecting sweat samples. The sampler wore two pairs of new gloves and full COVID-19 safety protection when collecting COVID-19 positive samples. COVID-19 negative samples did not require particular safety protection but the sampler wore a new pair of surgical gloves to prevent contaminating samples with her/his own odours. The same brands of gloves were used for COVID-19 positive and negative individuals from the same hospital. Before being sampled, each individual was told about the protocol and signed an informed consent form (Fig 1).

Axillary sweat samples were collected because it seems a promising substrate for canine detection [21, 44], is the key odour for search and rescue or tracking dogs [45], and the axillary region is easily accessible (Fig 2). Furthermore, this site is unlikely to be contaminated by the saliva of a COVID-19 positive patient. However, since it is not known whether sweat is a SARS-CoV-2 transmission route via skin-to-object-to-mucosa contact [46], careful manipulation of axillary sweat samples is required.

The sampling material used were 2x2-inch sterile gauze swabs used by the hospitals or sterile gauze filters used for explosive detection (provided by DiagNose comp.), and inert polymer tubes used for explosives, drugs or criminology detection (provided by Gextent comp.) (Fig 3). The polymer tubes can adsorb both polar and apolar molecules to try to identify potential SARS-CoV-2 sweat biomarkers. Two sampling materials (i.e., gauze or polymer tubes) were used because at the beginning of this proof-of-concept study, it was not known which of the two was the most effective. The sampling material remained in contact with the skin for 20 minutes. The average amount of sweat obtained is around 75 mg for both the gauze swab and the cellulosic filter.

Samples were stored in hospital anti-UV sterile containers, disinfected by the sampler's helper, coded (including left or right underarm), then placed into a second plastic envelope. Individual anonymous data were registered on a form for each coded sample (Fig 4). All samples were transferred from the sampling site to the testing site in separated coolers (of the same

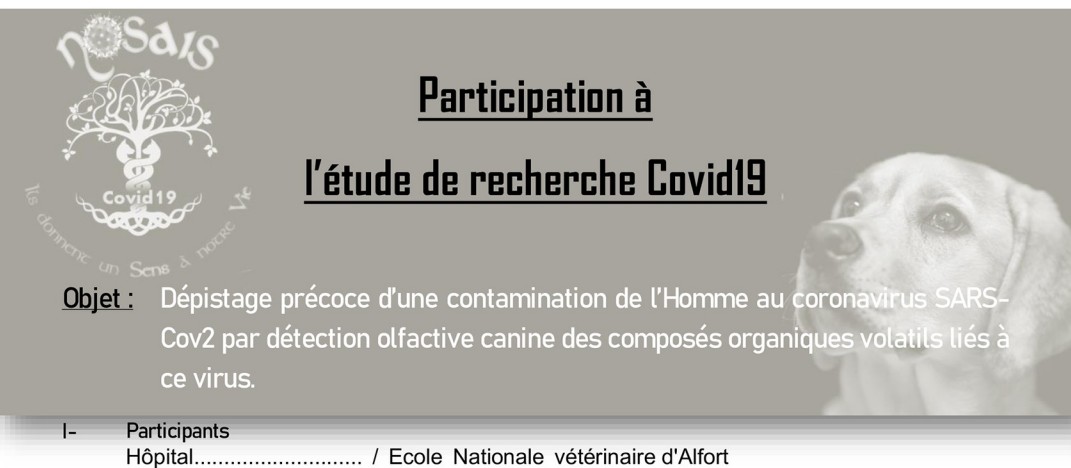

I– Participants

Hôpital........................... / Ecole Nationale vétérinaire d'Alfort

II– Introduction

La détection olfactive précoce de maladies chroniques prolifératives (cancers) ou dégénératives (Parkinson) par des chiens dûment formés se développe rapidement de par le monde (France : projets NOSAÏS et KDOG). Les recherches antérieures ont confirmé la présence de VOC (Volatile Organic Compound) spécifiques engendrées par l'infection virale chez les personnes infectées, tout comme c'est le cas pour certaines maladies virales des bovins maintenant détectées par des chiens aux Etats Unis (Par exemple, maladie des muqueuses).

L'objectif de ce travail de recherche est **d'augmenter nos moyens de dépistage facile et rapide**, ce qui sera précieux à l'avenir. Et c'est en acceptant aujourd'hui de vous faire prélever, que vous participerai de façon active à **ce projet de santé mondiale !** Nous insistons sur le mot mondial, car si nous réussissons, la méthodologie sera diffusée dans le monde entier.

Important : L'ensemble des résultats des tests demeureront codés et anonymes.

Le devenir de vos prélèvements ?

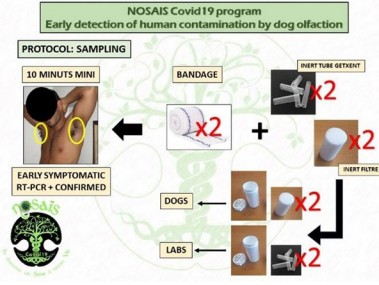

Perspectives :

La réussite de cette approche pourrait permettre de faire un pré-dépistage d'un grand nombre de personnes visant à ne tester de manière classique que les personnes positives à ce pré-test (gain de tests et de temps).

La validation de ce test pourrait présenter un grand intérêt lors de la procédure de déconfinement et en l'occasion de possibles vagues épidémiques futures.

Nom, prénom et signature du patient consentant, précédé de « lu et approuvé » : En Date du : / / .

**Fig 1. Individual informed consent form.**

brand) for COVID-19 positive and negative samples. Coolers were cleaned and disinfected with a 10% aqueous acetone solution after each use. All samples were then stored in the same place at the two sites (but positives and negatives were not mixed) at constant temperature (+18˚C for Alfort site and +6˚C for Beirut site) and humidity (45%).

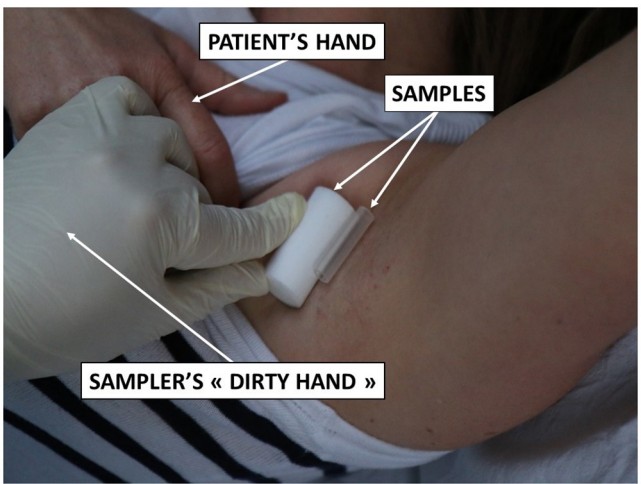

**Fig 2. Underarm sampling.**

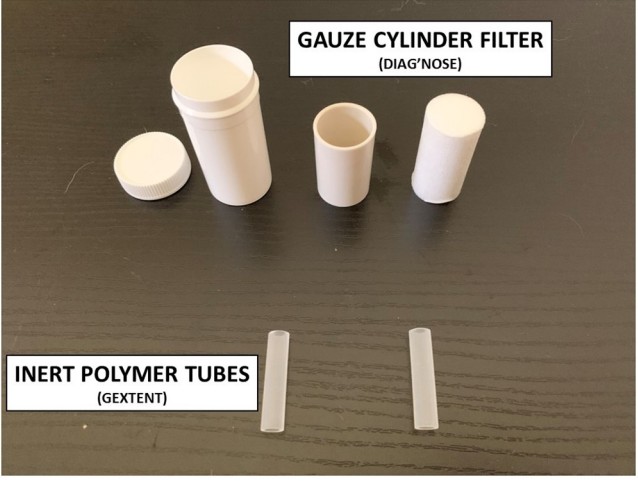

**Fig 3. Sampling materials.**

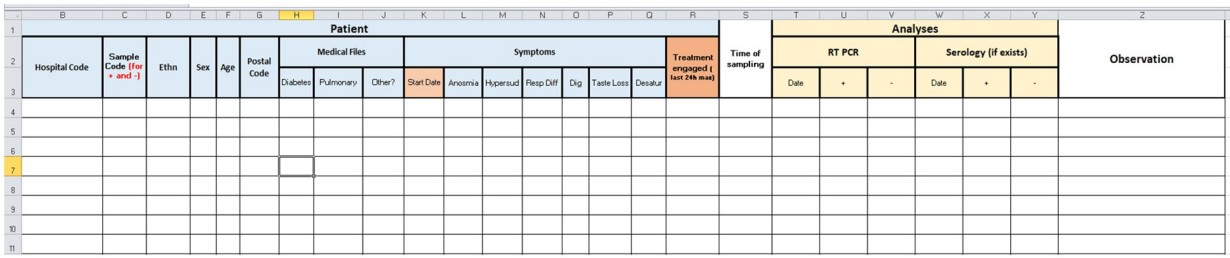

**Fig 4. Anonymous form for each coded sample.**

## Canine resources

Dogs trained for this proof-of-concept study were explosives detection dogs, search and rescue dogs, and colon cancer detection dogs. Explosives detection dogs are trained to detect 30 to 40 different types of explosives and work on a line of samples that they have to sniff individually. For such dogs, if COVID-19 positive samples have a specific odour, they only have to memorise this additional odour (and subsequently generalise it). Search and rescue dogs are trained to perform area searches, mainly using the victim's scent. Colon cancer detection dogs are trained on rectal gases. Drug detection dogs were not selected as candidate dogs since it cannot be ruled out that COVID-19 positive or negative individuals have used prohibited substances, the catabolites of which would be excreted into the sweat and subsequently detected by such dogs.

## Safety of the dogs regarding SARS-CoV-2 infection

There have been very few reports on dogs being passive carriers of SARS-CoV-2. The most synthetic and documented case concerned two dogs living in a SARS-CoV-2 infected household, out of a total of fifteen households [47]. The dogs were quarantined and remained asymptomatic. The authors concluded that it was unclear whether infected dogs could transmit the virus back to humans. In the USA, Idexx Laboratories tested more than 3,500 dogs, cats and horses from places where community transmission of SARS-CoV-2 was occurring in humans and found no positive animals [48]. A recent study, which is not yet published, provided some evidence of the absence of SARS-CoV-2 infection in dogs in close contact with a cluster of COVID-19 patients [49]. Finally, the CDC (Centre for Disease Control and Prevention, USA) attests that there is no evidence that pet animals, and especially dogs, play any significant role in SARS-CoV-2 transmission or spread [50].

It is assumed that like SARS-CoV, SARS-CoV-2 does not survive longer than a few hours on cotton [51]. So, for safety reasons, samples were not used for training or testing sessions within 24 hours of collection. A more recent study concludes that absorbent materials like cotton are safer than unabsorbent materials for protection from SARS-CoV infection [52].

## Training protocol

Fourteen dogs were trained to work on a line-up of cones, used as sample carriers (Figs 5 and 6), and to mark the cone containing the COVID-19 positive sample by sitting in front of it (Fig 7). The training method was based on positive reinforcement (the dog gets its toy for each correct marking). The whole training session was carried out according to the following 4-step procedure: (1) learning line-up work (all cones in the line-up are empty and the dog is rewarded with its toy when he sniffs each of the empty olfaction cones); (2) memorising the COVID-19 sample odour (the dog is rewarded when he marks the cone with the COVID-19 positive sample and all the remaining cones in the line-up are empty); (3) empty cones are replaced by mocks (i.e., cones containing the sample material only; the dog is rewarded when he marks the COVID-19 positive sample); (4) COVID-19 negative samples are introduced into the line-up which contains one COVID-19 positive sample, one or two mock(s), and COVID-19 negative sample(s). For each dog, memorising the specific COVID-19 positive sample odour required less than one day, with the dogs sniffing between four and ten COVID-19 positive samples. The handler judged when the dog was trained and ready for the testing session, based on the dog's behaviour.

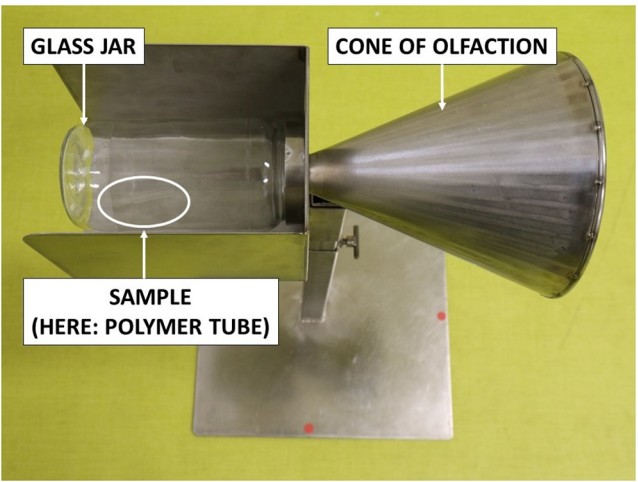

Fig 5. Testing equipment.

## Testing protocol

Once the dog was considered trained, the testing sessions started with new COVID-19 positive and negative samples (i.e., samples which were not used for the training sessions). During the testing sessions, each "trial" was conducted in a dedicated room containing 3 or 4 cones located at the back of the room (see Fig 6). The data recorder randomly placed one COVID-19 positive sample behind one cone, and at least one COVID-19 negative sample, and between zero and two mocks behind the remaining cones in the line-up. All the samples in the line-up were from the same hospital and were made of the same material (gauze or polymer tubes). The presence or absence of mocks depended on the way the dog usually worked with its handler for explosive detection, search and rescue, or for colon cancer detection. All the trials for one dog used the same total number of cones in the line-up (four cones for five dogs, and three cones for one dog). One sample (either COVID-19 positive or negative) could be used in two or three separate trials for one dog. As recommended by Johnen et al. [31], the location of the COVID-19 positive

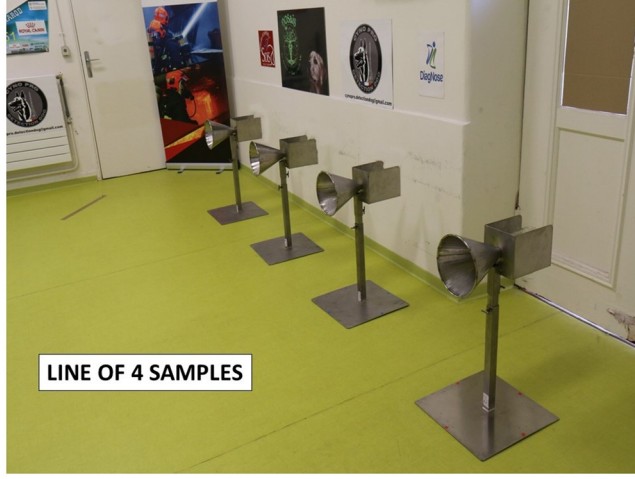

Fig 6. 4-olfactory cone line-up.

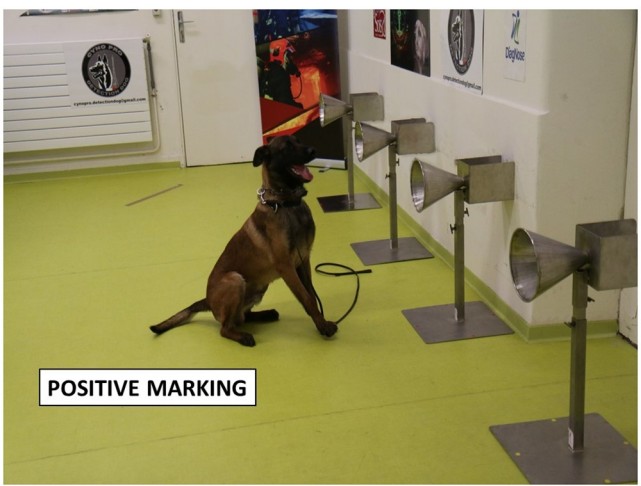

**Fig 7. A dog marking a cone on a 4-cone lineup.**

and negative samples in the line-up was randomly assigned by a dedicated website (http://www.randomization.com/), and was blinded to the dog and its handler. During the testing session, the data recorder knew the COVID-19 positive sample location but there was no visual contact between the data recorder and either the dog or the dog handler. Once the samples were placed behind the cones, the dog entered the room with its handler who asked the dog to sniff each cone in the line-up. Since the dog had been trained to mark the position of one, and only one, COVID-19 positive sample behind a cone, the dog was left free to sniff the cones before marking one cone. Once a cone was (correctly or incorrectly) marked, the trial was stopped. The data recorder indicated to the dog handler whether the dog correctly marked the cone or not; if correct, the dog handler rewarded the dog. The trial was considered a "success" if the dog correctly marked the cone with the COVID-19 positive sample, otherwise it was considered a "failure". After each trial, the dog and its handler left the room, and the data recorder placed new samples behind the cones in the line-up. The line-up in this protocol is "simultaneous" rather than "sequential" (i.e., each cone is sniffed by the dog and if the dog marks a cone, the handler asks the dog to resume the task for the remaining cones in the line-up).

The testing period lasted 21 days because most dogs could not work every day. During the training and the testing sessions, the standard environmental conditions for canine olfactory detection were observed (temperature of +18˚C and 50% humidity). No disease, symptoms or clinical abnormalities have been reported for any of the dogs involved in the study.

Both samples and cones were manipulated by the same person (the data recorder), wearing sanitary barrier protections and a pair of new surgical gloves at each trial, to avoid any olfactory contamination or interaction. All cones were cleaned with clean water after each trial, and twice daily were cleaned and disinfected with 10% aqueous acetone solution, and then dried.

## Statistical analysis

Since this is a proof-of-concept study, the initial objective was to provide evidence that a trained dog can detect one COVID-19 positive sample in one olfactory cone from a line-up of three to four cones, containing either mocks or COVID-19 negative samples. It was not possible to calculate the sensitivity, specificity or false alert rates (as recommended by Johnen et al. [31]) because the line-up was simultaneous not sequential. The study outcome focused on the success rate, which was calculated by dividing the number of successful trials by the number of

trials carried out by the dog. If, for one dog, the lower bound of the 95% confidence interval (calculated by using the Clopper-Pearson's method [53]) of the estimated success rate was higher than the success rate obtained by chance alone (thereafter called "random choice proportion"), then this result would provide some evidence that this dog could actually detect COVID-19 positive samples. The dogs were initially trained to discriminate between sweat samples and mocks before being trained to discriminate between COVID-19 positive and negative samples (step 3 of the training). Therefore, the random choice proportion was calculated by subtracting the number of mocks present in the line-up, then dividing 1 by the number of COVID-19 positive or negative sweat samples in the line-up. For example, a 4-cone line-up containing one mock would produce a random choice proportion of 33% (i.e., 4–1 = 3, 1/3 = 33%). To address the impact on the success rates of the dogs being presented the same COVID-19 positive sample several times, success rates are also shown according to whether it was the first, second, or third time the sample had been presented.

## Results

A total of 95 symptomatic COVID-19 positive individuals and 82 asymptomatic COVID-19 negative individuals were recruited for the study, producing 177 sweat samples for the use in the study trials. The Paris site study sample consisted of 27 symptomatic COVID-19 positive individuals and 34 asymptomatic COVID-19 negative individuals. The percentage of females was similar between the two groups (52% for COVID-19 positive and 56%, for COVID-19 negative individuals), but the mean age was higher in the COVID-19 positive group (70 years) than in the COVID-19 negative group (42 years). The Beirut site study sample consisted of 68 symptomatic COVID-19 positive individuals and 48 asymptomatic COVID-19 negative individuals. The percentage of females (43% and 46%, respectively) and the mean age (48 years and 42 years, respectively) were similar between the COVID-19 positive and COVID-19 negative groups.

Out of the 14 dogs which started training, eight dogs did not participate in testing because they were not considered ready at the time of the beginning of the testing session (which could not have been postponed). Six dogs participated in testing and their training period ranged from one to three weeks.

The characteristics of the six dogs are presented in Table 1. Five dogs were Belgian Malinois Shepherds because this is the most commonly used breed in French working dog organisations. Three dogs were explosive detection dogs (Guess, Maika, and Gun), two were colon cancer detection dogs (Bella and Jacky), and one was a search and rescue dog (Gun).

For three dogs (Oslo, Bella, and Jacky), the 4-cone line-ups contained one COVID-19 positive sample and three COVID-19 negative samples without any mocks. For one dog (Gun), most of the trials (44/47) were performed on 3-cone line-ups containing one COVID-19 positive sample and two COVID-19 negative samples without any mocks. For the last two dogs

**Table 1. Characteristics of the 6 dogs participating in the study testing sessions.**

|  | Name | Gender | Breed | Age | Organisation involved | Specialty |
|---|---|---|---|---|---|---|
| **Paris** | Guess | F | Belgian Malinois | 8 years | Cynopro detection Dog | Explosives |
|  | Maika | F | Belgian Malinois | 3 years | Fire and Emerg. Dept. 77 | Search and rescue |
|  | Gun | M | Belgian Malinois | 16 months | Cynopro detection Dog | Explosives |
|  | Oslo | M | Belgian Malinois | 18 months | DiagNose | Explosives |
| **Beirut** | Bella | F | Belgian Malinois | 6 years | Mario-K9 | Colon Cancer |
|  | Jacky | M | Jack Russell terrier | 3 years | Mario-K9 | Colon Cancer |

(Guess and Maika), the 4-cone line-ups contained one COVID-19 positive sample, at least one COVID-19 negative sample, and either one or two mocks.

Table 2 presents the results obtained from the six dogs, according to the number of mocks in the line-ups. The success rates ranged from 76% (for Maika completing 17 trials with 4-cone line-ups containing two mocks) and 100% (for Bella and Jacky completing 4-cone line-ups containing no mocks). There was one other 100% success rate but it was calculated from just three trials containing one mock in 3-cone line-ups (for Gun). In all but one situation, the lower bound of the 95% confidence interval (CI) of the success rates was higher than or equal to the random choice proportion. The exception was the 100% success rate calculated from the three trials with one mock completed by Gun, where the lower bound of the 95% CI was 29% and the random choice proportion was 50%.

Table 3 presents the success rates according to whether the COVID-19 positive samples were being presented for the first, second, or third time to the four Paris site dogs (the data were not available for the Beirut site). For Guess and Maika, the success rates were higher for trials involving COVID-19 positive samples presented to the dog for the first time (92% and 87%, respectively), compared to trials where the positive samples were presented for the second (75% and 77%, respectively) and the third (75% and 83%, respectively) times. Conversely, the success rates for Gun and Oslo were higher for trials involving COVID-19 positive samples presented for the second time (96% and 100%, respectively) compared to trials where positive samples were presented for the first time (83% and 90%). However, for the trials when Gun and Oslo were presented the COVID-19 positive samples for the first time, the lower bound of the 95% CI of the success rates was higher than the maximum value of the random choice proportion reached in our study (50%, see Table 2).

## Discussion

In a context where there is a lack of diagnostic tests available to perform mass detection of SARS-CoV-2 in many countries worldwide, exploring the possibility of using dog olfactory detection as a rapid, reliable and cheap "tool" to pre-screen people or perform rapid checking in certain circumstances is important. The first step for such an approach was to determinate if trained dogs can discriminate between axillary sweat samples from symptomatic COVID-19 positive patients and asymptomatic COVID-19 negative individuals.

The results of this proof-of-concept study are a promising first step providing some evidence that dogs may be able to detect COVID-19 positive samples collected from the axillary

**Table 2. Results obtained from the 6 dogs participating in the study testing sessions.**

|  | Number of mocks / number of cones in the line-ups* | Number of trials | Correct identifications | Success rate [95% CI] | Random choice proportion |
|---|---|---|---|---|---|
| Guess | 1/4 | 34 | 26 | 76% [59%-89%] | 33% |
|  | 2/4 | 18 | 17 | 94% [73%-100%] | 50% |
| Maika | 1/4 | 38 | 32 | 84% [69%-94%] | 33% |
|  | 2/4 | 17 | 13 | 76% [50%-93%] | 50% |
| Gun | 0/3 | 44 | 39 | 89% [75%-96%] | 33% |
|  | 1/3 | 3 | 3 | 100% [29%-100%] | 50% |
| Oslo | 0/4 | 31 | 29 | 94% [79%-99%] | 25% |
| Bella | 0/4 | 68 | 68 | 100% [95%-100%] | 25% |
| Jacky | 0/4 | 68 | 68 | 100% [95%-100%] | 25% |

CI: Confidence interval;

* All line-ups contained one COVID-19 positive sample, and at least one COVID-19 negative sample.

Table 3. Results obtained from the Paris site dogs according to first, second, or third presentation of the COVID-19 positive samples.

| | First, second, or third presentation of the COVID-19 positive sample | Number of trials | Correct identifications | Success rate [95% CI] |
|---|---|---|---|---|
| Guess | 1st | 24 | 22 | 92% [73%-99%] |
| | 2nd | 24 | 18 | 75% [53%-90%] |
| | 3rd | 4 | 3 | 75% [19%-99%] |
| Maika | 1st | 23 | 20 | 87% [66%-97%] |
| | 2nd | 26 | 20 | 77% [56%-91%] |
| | 3rd | 6 | 5 | 83% [36%-100%] |
| Gun | 1st | 24 | 20 | 83% [63%-95%] |
| | 2nd | 23 | 22 | 96% [78%-100%] |
| Oslo | 1st | 21 | 19 | 90% [70%-99%] |
| | 2nd | 10 | 10 | 100% [69%-100%] |

CI: Confidence interval.

sweat of individuals showing clinical COVID-19 symptoms and who are positive on SARS-CoV-2 RT-PCR or PCR tests.

During the testing sessions, two samples collected from *a priori* COVID-19 negative individuals (i.e., individuals who met the inclusion criteria to be recruited as controls) were repeatedly marked by two dogs. The information was immediately sent to the concerned hospital and PCRs were performed again on these two individuals who turned out to be positive. These individuals and their corresponding samples, as well as the trials which included these two samples, were excluded from the data since, in these cases, the line-up contained more than one COVID-19 positive sample. False negative RT-PCR results are not uncommon [54–56], but these two events support our working hypothesis. Overall, our results are in accordance with a recent pilot study which investigated the ability of trained dogs to detect saliva or tracheobronchial secretion samples from symptomatic COVID-19 positive patients [32].

By filming each trial for each dog, we were able to understand why some trials were unsuccessful (i.e., a dog marked a COVID-19 negative sample or a mock). For example, a horse walking close to the testing room (on the site at the Alfort veterinary school), or people not respecting the protocol and making too much noise close to the testing room. However, there were other unsuccessful trials which could not be explained. This observation is not as negative as it first appears. Even if trained dogs are able to correctly discriminate symptomatic COVID-19 positive individuals from asymptomatic COVID-19 negative ones, they should not be considered a perfect diagnostic method, but rather a complementary tool in settings where diagnostic tests are not readily available or while diagnostic tests do not have high levels of accuracy [57].

Mocks were included in the line-ups for the study dogs who are accustomed to working with mocks for explosive detection or search and rescue. However, because the trial may have been easier for the dog when a line-up contained at least one mock compared to a line-up not containing any mocks, it was necessary to remove these mocks when calculating the random choice proportion. The lower bound of the 95% CI of the estimated success rates were all higher than or equal to the random choice proportion, except for one success rate. This success rate was 100% but was calculated from just three trials. We therefore cannot rule out that we may have lacked some statistical power. Although mocks are necessary to train the dogs, and can be taken into account in statistical analyses, future validation study testing protocols should not include mocks in their line-ups.

Our proof-of-concept study protocol used a simultaneous line-up rather than a sequential one. This therefore meant it was not possible to calculate the true positive and false negative

rates, as recommended by Concha et al. in 2014 [58], or sensitivity and specificity, as recommended by Johnen et al. [31]. Success rates were instead provided in this present study because each trial used a line-up containing just one COVID-19 positive sample, the rest being either mocks or COVID-19 negative samples. Providing a success rate for detection dogs is however considered an acceptable option by Johnen et al. [31]. This study was a necessary first step before conducting subsequent studies with sequential line-ups in which sensitivities and specificities will be calculated.

One other limitation of our study is the repeated use of some samples during the testing sessions. In this situation, we cannot rule out that the dog memorised the odour of a COVID-positive sample and marked it the second time because it had been presented before (dogs can memorise at least 10 odors [59]). However, in our study, the success rates did not seem to depend on whether the dog was being presented the COVID-19 positive sample for the first, second, or third time. For two out of the four dogs at the Paris site, the success rates were higher when the trials involved COVID-19 positive samples presented for the first time. For the other two dogs, the success rates were higher for the second presentation but were still above 80% for the trials when the COVID-19 positive samples were presented for the first time. These results support the hypothesis that olfactory memory did not play a major role, if any, in the discrimination task in our study.

The comparability criterion (i.e., positive samples are comparable to negative samples, except for the disease status) is recommended for detection dog studies [4]. If this criterion is not met, confounding bias may occur [5]. The criterion was met for the "hospital" factor, since a COVID-19 positive sample was collected in the same hospital as a COVID-19 negative sample but it may be possible that COVID-19 positive patient rooms had a different odour to COVID-19 negative patient rooms. However, as Walczak et al. wrote in their paper published in 2012, "hospital rooms may have a hospital odour, derived mainly from disinfectants, which may be a common component of all samples taken from donors who are inside a hospital." [43]. Therefore, in our data, there should not be confounding bias due the hospital odour.

Although the proportions of females were similar between COVID-19 positive individuals and COVID-19 negative individuals, both at the Paris and Beirut sites, the comparability criterion was not met for age at the Paris site (COVID-19 positive individuals were older than the negative ones). This lack of comparability is a strong limitation in this study. Body odour differs with age [60, 61] and humans seem to be able to discriminate age based on body odour alone [62]. This lack of age comparability at the Paris site may have introduced some noise that possibly altered the dog's performance, since the French dogs had lower success rates than the Lebanese dogs. Furthermore, it is possible that the French detection dogs detected odours excreted by the elderly because of their age, and not because of the COVID-19 positive status, so age then becomes a confounding factor [30]. However, the observed results from the Lebanese dogs (a success rate of 100% for the two dogs), where mean age was similar between COVID-19 positive and negative groups, do not support this latter hypothesis.

Information about medical conditions or medication use for recruited individuals was not systematically collected, so it was impossible to check the comparability criterion was met for these characteristics. COVID-19 negative individuals in our study were hospitalised for another reason and were probably on medication which differed from the ones used by the COVID-19 positive individuals. This situation may have impaired the dogs' performance or introduced some confounding bias. However, when some conditions (including diabetes, arterial hypertension, respiratory disorders, and hypothyroidism) were recorded for some included individuals, and for which the comparability criterion was not met, the dogs rarely marked COVID-19 negative individuals suffering from one of these conditions (data not

shown). We cannot generalise this statement to other disease conditions which were not recorded in our data.

Besides medical conditions or medication use which may differ between COVID-19 positive and negative individuals, the amount of sweat collected in the samples was not individually recorded and we cannot exclude that the quantity of sweat collected from COVID-19 positive individuals was higher than that from COVID-19 negative individuals. This may have been a confounding factor. However, the sampling procedure described in our study led to an inter-individual variability in sweat quantities not related to sample status (COVID-19 positive or negative), but rather related to individual characteristics.

Two sampling materials (sterile gauze swabs or filters, and inert polymers tubes) were used because at the beginning of this proof-of-concept study, it was not known which sampling material was the most effective. The sampling material should however not play a role as a confounder because the same sampling material was used within each trial. Therefore, the dog cannot have been influenced by the sampling material odour when trying to detect the COVID-19 positive sample.

Due to the aforementioned limitations, more research in this field is needed with studies which meet all the recommended criteria [4, 30, 31]. However, despite these limitations, our study has some strengths in accordance with previous recommendation [4, 30, 31]. These strengths included the fact that samples used for the training sessions were different from those used for the testing sessions, six dogs were used in this study which is considered a large number when compared to other detection dog studies, only one sweat sample was collected per individual and a large number of individuals were recruited in our study (n = 177), the sample position was randomised, and the dog and its handler were blinded to the sample locations. Although the data recorder was not blinded to the sample location, he remained at the back of the room and could not be seen by either the dog handler or the dog during the session. This situation reduced the chance of the data recorder directly influencing the dog's behaviour, but some unintentional influence on the choice of samples cannot be totally ruled out. Furthermore, this single blinding allowed the data recorder to tell the handler when the dog correctly marked a cone so the dog could be rewarded as soon as it corrected marked the cone. It should however be noted that the study should ideally be double-blind (i.e., both the handler and the data recorder are blinded to the sample positions), while being able to reward the dog using various methods [30].

The training period for the six study dogs (between one and three weeks) was much shorter than reported in some detection dogs studies. One reason for this is that eight out of the 14 dogs which initially participated in the training sessions were not considered ready by their handler at the time of the beginning of the testing sessions (which could not have been postponed). In operational settings, it is recommended to expect around six weeks to train a COVID-19 detection dog. Furthermore, this training time is in accordance with the study by Jendrny et al. on COVID-19 detection dogs [32]. The short training period can also be explained by the fact that the study dogs were already well-trained detection dogs. The explosives detection dogs are trained to detect between 30 and 40 different types of explosives and are accustomed to working on a line of samples that they have to sniff individually. For such dogs, if COVID-19 positive samples have a specific odour, they only have to memorise one additional odour. Search and rescue dogs are trained to perform area searches and work using the scent of the sweat. Although these dogs are trained to detect general human odour, without focusing on a person's specific odour (e.g. odour(s) related to a disease), they can easily be retrained to work in a scent line-up and discriminate odours. We also used colon cancer detection dogs which are trained on rectal gases. These detection dogs are accustomed to working with samples on line-ups and they quickly memorise a new odour.

The next step would be to carry out a double-blind validation study involving as many COVID-19 positive and negative individuals as this proof-of-concept study, which will provide the sensitivity and specificity of the dogs' ability to discriminate between COVID-19 positive and negative samples. Ideally, in this validation study, samples should be received from an independent source (e.g. from another organisation or even country) which have not been processed by the research team before being presented to the dogs. If the sensitivity and specificity are high enough, then this validation study will provide some evidence that national authorities could use trained COVID-19 detection dogs in settings where equipment and/or money are lacking to perform standard serology or RT-PCR tests, or as a complementary method in other settings.

## Supporting information

**S1 Data.**
(XLSX)

## Author Contributions

**Conceptualization:** Dominique Grandjean, Riad Sarkis, Bruno Maestracci, Pascal Morvan, Jean-Pierre Tourtier.

**Data curation:** Clothilde Lecoq-Julien, Aymeric Benard, Vinciane Roger, Eric Levesque, Eric Bernes-Luciani, Eric Gully, David Berceau-Falancourt, Joaquin Cabrera, Quentin Muzzin, Jean-Marie Broc, Anthony Lichaa, Georges Moujaes, Michele Saliba, Aurore Kuhn, Mathilde Galey, Benoit Berthail, Lucien Lapeyre, Anthoni Capelli, Aymeric Stainmesse, Erwan Etienne, Sébastien Voeltzel, Sofiane Mansouri, Aimé Dami, Lary Charlet, Mario Issa, Christophe Billy.

**Formal analysis:** Dominique Grandjean, Vinciane Roger, Quentin Muzzin, Capucine Gallet, Jean-Pierre Tourtier, Loïc Desquilbet.

**Funding acquisition:** Bruno Maestracci.

**Investigation:** Dominique Grandjean, Riad Sarkis, Clothilde Lecoq-Julien, Aymeric Benard, Eric Gully, David Berceau-Falancourt, Joaquin Cabrera, Capucine Gallet, Steevens Renault, Karim Bachir, Anthony Kovinger, Eric Comas, Aymeric Stainmesse, Erwan Etienne, Sébastien Voeltzel, Sofiane Mansouri, Marlène Berceau-Falancourt, Mario Issa.

**Methodology:** Dominique Grandjean, Riad Sarkis, Aymeric Benard, Pascal Morvan, David Berceau-Falancourt, Anthoni Capelli, Steevens Renault, Karim Bachir, Lary Charlet, Eric Ruau, Mario Issa, Jean-Pierre Tourtier, Loïc Desquilbet.

**Project administration:** Clothilde Lecoq-Julien, Aymeric Benard, Eric Bernes-Luciani, Bruno Maestracci, Pierre Haufstater, Hélène Bacqué.

**Resources:** Dominique Grandjean, Clothilde Lecoq-Julien, Aymeric Benard, Vinciane Roger, Eric Levesque, Eric Gully, David Berceau-Falancourt, Pierre Haufstater, Gregory Herin, Joaquin Cabrera, Capucine Gallet, Jean-Marie Broc, Leo Thomas, Anthony Lichaa, Georges Moujaes, Michele Saliba, Aurore Kuhn, Mathilde Galey, Benoit Berthail, Lucien Lapeyre, Anthoni Capelli, Anthony Kovinger, Eric Comas, Marlène Berceau-Falancourt, Lary Charlet, Eric Ruau, Carine Grenet, Christophe Billy, Jean-Pierre Tourtier.

**Software:** Quentin Muzzin.

**Supervision:** Dominique Grandjean, Riad Sarkis, Clothilde Lecoq-Julien, Aymeric Benard, Anthoni Capelli, Marlène Berceau-Falancourt.

**Validation:** Riad Sarkis, Clothilde Lecoq-Julien, Pascal Morvan, Steevens Renault, Karim Bachir, Aymeric Stainmesse, Erwan Etienne, Sébastien Voeltzel, Marlène Berceau-Falancourt, Aimé Dami, Mario Issa, Jean-Pierre Tourtier.

**Visualization:** Capucine Gallet, Hélène Bacqué, Karim Bachir, Erwan Etienne, Sébastien Voeltzel, Marlène Berceau-Falancourt, Eric Ruau.

**Writing – original draft:** Dominique Grandjean, Jean-Pierre Tourtier.

**Writing – review & editing:** Dominique Grandjean, Loïc Desquilbet.

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
