## [Decision Letter · Decision Letter 0]

13 Jul 2020

PONE-D-20-17256

Detection dogs as a help in the detection of COVID-19

Can the dog alert on COVID-19 positive persons by sniffing axillary sweat samples ?

Proof-of concept study

PLOS ONE

Dear Dr. grandjean,

Thank you for submitting your manuscript to PLOS ONE. After careful consideration, we feel that it has merit but does not fully meet PLOS ONE’s publication criteria as it currently stands. Therefore, we invite you to submit a revised version of the manuscript that addresses the points raised during the review process.

As you will read the 3 treviewers raised relatively similar methodological options and my opinion is converging in this direction.

To summarize as suggested by one of the reviewer you have two options:

- Additional data is added, providing the results of a double-blind test using samples from new people that have not previously been screened by the dogs, or

- The current abstract, discussion and conclusions are revised to provide a more critical appraisal of the results, covering possible limitations of the study and whether there are any other possible explanations for the dogs' detection performance and possible sources of unintentional bias.

It seems also clear that your manuscript would benefit from being corrected by a native english speaker.

We look forward to receiving your revised manuscript.

Kind regards,

Nadine Ravel

Academic Editor

PLOS ONE

Journal Requirements:

"The authors received no specific funding for this work"

We note that one or more of the authors are employed by a commercial company: "DiagNose, Cynopro Detection Dogs and Biodesive SAS"

Reviewers' comments:

Reviewer's Responses to Questions

**Comments to the Author**

1. Is the manuscript technically sound, and do the data support the conclusions?

Reviewer #1: Partly

Reviewer #2: Partly

Reviewer #3: Partly

2. Has the statistical analysis been performed appropriately and rigorously? 

Reviewer #1: No

Reviewer #2: No

Reviewer #3: I Don't Know

3. Have the authors made all data underlying the findings in their manuscript fully available?

Reviewer #1: No

Reviewer #2: No

Reviewer #3: No

4. Is the manuscript presented in an intelligible fashion and written in standard English?

Reviewer #1: No

Reviewer #2: Yes

Reviewer #3: No

5. Review Comments to the Author

Reviewer #1: This article investigates whether dogs can differentiate the axillary odor of individuals infected by SARS-CoV-2 from individuals that are not infected. The study is very well justified and potential applications for the management of the current pandemic (especially in case of a feared second wave) are highly relevant.

I however have some major issues with this study, as detailed below.

1) Design. With the current information provided in the method, the reader cannot be convinced that the authors controlled for a number of confounding factors. Namely, it should be made clear that the negative and positive samples cannot be different from each other based on other factors than infection to SARS-CoV-2. It is known that body odor changes according to medication intake, gender, age, health status, hormonal status etc. With this in mind, a number of questions can be raised here. Who are the COVID-negative patients: do they have other pathologies? If yes, which ones? Do they all have the same pathology, or different ones? Are they on medication? Are the COVID+ on medication and if yes, is it the same one for everybody? How are the negative samples chosen with regards to gender and age, according to the target (COVID+ sample) characteristics? Two odor sampling methods are mentioned (gauze swabs or filters and polymer tubes): Are they both used for negative and positive samples, and is the same sampling support used in the whole line during a trial? In sum, more information is expected in the method section on these aspects; the limits related to the possible confounding factors must be discussed in more details than it currently is; and the conclusion will probably need to be more cautious (as a consequence).

2) Statistical analyses. The authors mention accuracy of the dog’s responses in introduction and conclusion, in terms of sensitivity and specificity. It should be clearly defined, especially with regards of the tests performed in the present article. The signal detection theory is evoked only in discussion, and false positives are discussed in that same section. It is therefore expected to see the current data analyzed using these criteria, which is not the case.

3) Paragraph about human pheromones in the discussion (lines 359-382). I suggest to simply remove this paragraph, which is 1) erroneous (no pheromones have ever been identified in humans, and the androgenous compounds you are referring to are male – not female – compounds supposed to attract females), 2) very speculative (how did you estimate the fertile phase of the donor menstrual cycle?), 3) disproportionally long compared to other points to discuss that are more crucial and more directly related to the main question.

Minor points.

1) Please describe what the mocks are, and how many are there in the line.

2) Please better explain how the training was conducted with regards to sample type: are there negatives at this stage? Mocks?

3) Please explain body odor sample management. From what I understand, they are used fresh (minimum 24 hrs between odor collection and training). Are they thrown away after 3 trials? Are they frozen at any time in the process? Refrozen?

Finally, on a more formal aspect, the article should be proofread for English. Also, improvement of the writing would be welcome. Some sentences are difficult to understand (i.e., lines 237-240 / 395-398) and some others do not integrate well in the chain of arguments (i.e. lines 62-64 / 343-345). Most importantly, in its current form the article is very fractionated (“bullet points” feeling in the whole document): It would very much gain in fluidity and articulation of the ideas if the authors work on this aspect.

Reviewer #2: Review of the manuscript PONE-D-20-17256

The idea of using trained dogs for COVID-19 screening was boosted by mass media in many countries e.g. by the BBC in UK, and by announcing by many researchers and dog trainers, that they intend to train dogs to detect COVID-19. It is obvious that first published experimental works in this field, will have a great potential for being broadly cited or comercialized. However, it could be presumed that such experiments and manuscripts are probably accomplished under a strong pressure of time, to be published as the first. Therefore the papers like the present one, which, to my knowledge is the first paper on this topic, should be reviewed and evaluated particularly causciously and critically. It is important not to disseminate facts that are not based on sound scientific experiments. This is particularly important since the reviewed paper is already available online at https://www.biorxiv.org/content/10.1101/2020.06.03.132134v1.full.pdf as not certified by peer review.

For justification of their study the authors cited publications on cancer detecting dogs and dogs alerting hypoglycaemia and seisure in human patients. However, it should be mentioned that despite of about 20 publications within the last 17 years, the medical detection dogs have not been applied for practical clinical screening so far. This was evident on the latest international conference on cancer detection by dogs, organized by the Curie-Institute in Paris in October 2019. Also, despite of numerous chemical studies within more than 25 years, using gas chromatography / mass spectrometry, which found some differences in volatile organic compounds (VOCs) between cancer positive and cancer negative patients, no single VOC or a simple combination of VOCs were identified, that could be reliable cancer marker(s), mature for practical cancer screening. The reason for problematic application of methods based on identivication of VOCs for disease screening, either by chemical analyses or by trained animals, is a great variability of emited VOCs due to day-to-day differences in diet, medication and numerous other confounding factors.

The aim of the reviewed study was to assess if dogs could be trained to discriminate body odor of patients infected with SARS-Cov-2, from body odor of healthy controls.

Such aim is justified by the current and probably future situation with COVID-19, untill an effective and commonly accessible COVID-19 vaccine is available.

The experiments were conducted at 3 different locations, with a total of 18 dogs which were previously trained either for explosives detection or for search and rescue or for cancer detection.

The Authors collected altogether 101 sweat samples from patients showing clinical symptoms of COVID-19, being also COVID-19 positive on RT-PCR or PCR test for SARS-CoV-2. Negative samples (n=97) were collected the same way, by the same trained staffs, from patients who have no clinical symptoms related to COVID-19, and were negative on SARS-CoV-2 PCR test.

The main flaw of collecting sweat samples for this study was, that the samples were not controlled quantitatively for the amount of the sweat. It is known that one of the typical COVID symptoms is a high fever. Humans with fever sweat more intensively than healthy people, so the amount of the sweat on the sample may be a clear cue for dogs.

As an example, our early studies in the eighties, on dogs detecting so called „silent estrus” in cows, could be mentioned. We used cotton swabs for collection vaginal mucus from cows in estrus vs diestrus. Our dogs perfectly indicated estrus samples. Only later we noticed that cows in estrus usually secreted more mucus and this was a cue for the dogs. When the dogs became more „experienced” they perfectly indicated cotton swabs moistured even with water. This made me causcious and critical about dog’s reliability. For dogs, odors like cow’s in estrus, human cancer, drugs or explosives, or SARS-CoV-2, play no biological role and such odors are indicated by dogs only because an operant conditioned response has been trained by producing association between odors and a reward in form a treat or favorite object to be retrieved and play with. Therefore, the trained dogs, especially when they are more „experienced”, will seek for a cue or a simple solution, to earn a reward. Such strategy of dogs may produce frequent false alerts. At identification of human scents by dogs it should be remembered that samples of human scent contain both individual component and other components as those related to diet, disease(s), medication, hygiene, cosmetics, background odors of the location as well as own odors of the sampling materials (odor carriers).

In lines 346-349 the authors report on two samples that were negative according to inclusion criteria for negative samples, but were indicated by two dogs as positive and the PCR re-test showed that in fact they were positive. This, however, is not a convincing evidence that the dogs sniffed out specific molecules induced by the SARS-CoV-2 virus presence. These two persons were not controlled for the amount of the sweat or body temperature measured precisely at the time of sample collection. Without knowing if the odor samples differ quantitatively in the amount of the sweat, it is not clear whether the dogs indicated a hypothetic, characteristic odor emited by COVID positive patients, or rather indicated samples from people with fever, who sweat more.

The Authors mentioned that the odor samples were collected „ at the same location”. This is methodologically sound but it should be precised what does it means „the same location” in this context ? Was it the same room or the same buiding or the same district ? The authors must be more precise here, because it is known that the dogs might be conditioned on the characteristic „hospital odors”, as shown by Walczak et al.(2012) in cancer detecting dogs, when the cancer positive samples were collected mostly in hospitals.

Another problem with COVID-positive sweat samples is that positive patients are staying either in infectious hospitals or in isolation wards that are probably intensively desinfected.

In addition, the access of healthy COVID-negative healthy donors (controls) to the same room where sick patients are staying, is hazardeous and strictly limited. This may be a serious constraint for collecting samples for further studies by other researchers.

The description of the training methods only in 3 lines is very scarce. Were the dogs rewarded in every training and testing trial and who and how gave a signal that the dog’s indication was correct ?

Lines 219-220 - It is true that search and rescue dogs (SRD), trained to perform disaster and area search, mainly work through the scent of the sweat. However, it must be taken into consideration that SRD are trained for detection of a generalized human odor, without discriminating individual persons and persons’ specific odor e.g. odor(s) related to a disease. Such dogs are relatively easy to be re-trained to work in a scent lineup to discriminate odours of particular person, but a re-training, to be reliable, takes longer than 1-4 hours and 4-10 trials.

In lines 262-264 the Authors stated: „…For each involved dog, the acquisition of the specific odour of COVIS-19+ sweat samples required from one to four hours, with an amount of positive samples sniffing ranging from four to ten….”

I am skeptical about the ability of each dog to learn precisely the task within such a short time. For example for cancer detecting dogs, the training period according to 13 papers published in the period 2003-2014, ranged from several weeks (Pickel et al. 2004) to 16 months (Cornu et al.2010), and on average took 8 months.

Even if the Authors could observe that their dogs correctly indicated the target samples after one hour, performing only four trials from being confronted for the first time with COVID sweat samples, it seems inredible that dogs could acquire the ability to detect COVID odor so quickly. It can not be excluded that the dogs indicated samples in the lineup, that are merely different from the others e,g, by containing more sweat.

In lines 265-267 or in lines 294-296 the Authors did not precise what was a criterion (percentage of correct indications, false alerts, misses, regularity of sniffing the samples in the lineup ?) for considering the dog to be fully trained and ready for testing.

From the lines 294-296 it could be concluded that separate samples were used for the training and testing, which is methodically correct. However, there is no information how many positive and negative or mock samples, out of the totals given in Table 1, were used for the trainig and real testing.

The authors did not conduct so called „zero trial” with no target samples in the lineup. This is another flaw which could have a consequence by making the overall conclusion in lines 407-408 premature. Although negative and mock samples were used in the lineup (lines 269-270), but there is no explanation what kinds of mocks were used. Were they samples with more sweat ? or samples from donors with fever but not related to COVID-19 (for example from people with seasonal influenza) ? „Zero trials” usually provoke more false alerts, especially on samples that are simple different from the others.

Minor remarks:

The manuscript shoud be better structured. For example the major portion of the Conclusion is appropriate rather for the Discussion section, whereas the real conclusion is given as the last sentence or the Discussion (in lines 407-408)

Line 64 More original papers should be cited instead of a handbook.

Lines 76-78 „…Scent detection by animals has been published in a large number of diagnostic studies, which all suggest similar or even superior accuracy compared with standard diagnostic methods…” – nevertheless, the scent detection by animals is not used for medical diagnoses or screenings on a large scale (probably except for tuberculosis detecting Gambian pouched rats in Africa). The medical diagnosis of a disease should not be confused with screening. Scent detection by animals will never be a diagnosis.

Line 83 „…The first clinical investigation of cancer was published by Willis on bladder cancer in 2010, after having published a proof of principle study in 2004…” – this information is only partly true: McCulloch et al. published their study on canine detection of lung and breast cancer in 2006.

Lines 88-89 „….The dog’s nose is also now currently utilised in order to prevent the consequences of crisis (or even prevent them) for diabetic [30, 31, 32, 33] and epileptic people [70]….” - The accuracy and reliability of hypoglycaemia and seizure alerting dogs, when considering on a daily basis, is still questionable. The alerting dogs rather contribute to feeling more safe and thus to improving the quality of life of the patients, than to precise alerting the hypoglycaemia or seizure episodes.

Lines 115-117 „….In a second time……” – this assumption is not quite clearly formulated in English.

Lines 154-158 Why two types of sampling supports are used ?:

(1) - Sterile gauze swabs utilised by the hospitals or sterile gauze filters utilised for explosive detection, and (2)- Inert polymers tubes utilised for explosives, drugs or criminology detection)

Line 289 Why acetone was used for disinfection of sample carriers ? Acetone is an irritant, causing mild skin irritation and moderate to severe eye irritation which may be repelling for dogs.

Lines 292-297 „…A failure was defined if the dog alerts on any of the other boxes. Trials were registered from the moment dogs recognized positive sample among other samples (mock or negative ones) without having sniffed any of them previously to the test. If the dog marks first a negative sample, the trial is considered as failed, even if he marks the positive afterwards….”

- the authors did not precise whether the dogs have to sniff ALL samples prior to the indicatication. Have the dogs to sniff the remaining samples after the correct indication? If so, would the trial be declared as a failure, if the dog made a false alert AFTER prior correct indication in the same trial? . This makes that the calculation of correct indication by chance should take into account how many samples a dog actually has sniffed. Therefore, when the dog indicates correctly for example on the first sample in the lineup and does not sniff the remaining samples, the probability of correct indication by chance at 7 samples in a lineup is not 14% but 50%. This would mean that the probability of correct indication by chance is varying from trial to trial ?

Line 350 „….dogs have no reason to lie….” – this is too colloquially expressed. With regards to odor detecting dogs one should not speak about "lying" or "telling truth".

From the dog's perspective, detecting trained odors as for example the hypothetical COVID-19 odor, that have no biological relevance, is a rather a game to earn a reward. Dogs, even if tey are well trained, when they are getting more routined, will try to get a reward at a simple way, which means that they will try to find any additional cues to indicate the target odor in the lineup. Therefore the dogs' performance in the scent lineup has to be permamently controlled throughout the whole deployment period, because it may vary. This seems to be one of the reason why so called medical detection dogs have not been used in medical screening on a regular basis.

Table 3. The terms detection sensitivity and specificity, which derive from the point 1-4 in lines 385-388, should be given, or percentage of false alerts should be given instead of the column with the number of correct indications, which is redundant in this table

The English of the manuscript is for me as a non-native speaker understandable but suffers from many language and wording imperfections. It should be edited by a native English speaker, who also has some knowledge in the specific study field.

Line 48 „..State of play..” ? – rather „State-of-the-art.”

Line 51 What does it means „..for those of interest in dog’s olfaction…” in this context?

Line 148. „…In order to avoid any dog creancement (?) on the olfactive “background noise

In several places of the manuscript there are statements that are difficult to follow

In lines 115-117 the authors write: „…In a second time, we attempt to validate the method with the dogs checking samples one by one instead of comparatively, in order to fit with what could be an operational utilisation of such trained dogs...” – what does it means exactly

Line 372 „ In Sheakspeare’s day…” ? - rather is „ in Sheakspeare’s era” or „in Sheakspeare’s time”

Lines 372-374 the passage with the anecdote on peeled apple under woman’s arm should be deleted because it is too colloquial and not closely related to the reviewed study.

Line 404 „ …cynotechnicians…” ? or dog handlers ?

Line 455 „…by small…” ? – should be „ by smell”

In conclusion, this first report on dogs trained to detect COVID-19 is interesting but a flaw in the methodology of collecting scent samples (no control for the amount of the sweat) makes that the conclusion that „…there is a high evidence that dogs can detect a person infected by the virus responsible for COVID-19 disease …” seems to be premature.

In addition, some language imperfections make me to evaluate the manuscript as not acceptable for publication in PloS ONE in the present form and needing a major revision before resubmission.

First of all, in the major revision the issue of the amount of sweat in odor samples should be addressed, including replication of the trials.

Reviewer #3: General comments

Thank you for conducting and reporting this very interesting study. In my opinion, this study is a good first step towards testing whether dogs can detect odours associated with SARS-CoV-2 infection. You have used a reasonable sample size of dogs and SARS-CoV-2 samples and the methods are generally appropriate. A main difficulty with any study of this type, where it is not known whether the odour the dogs are being trained to detect exists (in this case SARS-CoV-2 related), is demonstrating beyond reasonable doubt that dogs are detecting the odour rather than incidental contamination, and demonstrating robustly that the research team is not unintentionally influencing/cuing the dogs. Sources of odour contamination may include, for example, medication used in SARS-CoV-2 patients, hospital odours, differences in the odour of the storage containers of positive and negative samples and different odours arising from how positive and negative samples are handled. You have taken some steps to try to control for these issues. However, in my opinion, it is not possible to conclusively say that the sweat of SARS-CoV-2 contaminated persons can be detected by trained dogs based on the methods and results you present so far. In places, the methodology and controls you used just need further clarification in order to rule out possible sources of bias and alternative explanations that the dogs may be detecting contaminating odours associated with the SARS-CoV-2 samples.

Most importantly, to support the conclusion that the sweat of SARS-CoV-2 contaminated persons can be detected by trained dogs, a more rigourous, double-blind test is required, in which the research team is provided with new samples from an independent source which they are asked to assess using the dogs. The dog handlers, sample handlers and data recorders (everyone involved in assessing the samples) should be blind to the sample identities (positive or negative) until the dogs have conducted the test. Methods for how to do this have been presented in a number of previous medical detection dog studies.

I would suggest two options for revising the paper:

- Additional data is added, providing the results of a double-blind test using samples from new people that have not previously been screened by the dogs, or

- The current abstract, discussion and conclusions are revised to provide a more critical appraisal of the results, covering possible limitations of the study and whether there are any other possible explanations for the dogs' detection performance and possible sources of unintentional bias.

Potential sources of bias/limitations in medical detection dogs studies have been summarised previously (e.g. Elliker et al 2014. Key considerations for the experimental training and evaluation of cancer odour detection dogs; Edwards et al 2017 Animal olfactory detection of human diseases: Guidelines and systematic review; Johnen D, 2017. An approach to identify bias in scent detection dog testing). I recommend you review these guidelines and report where your study meets these criteria, and/or discuss limitations where your study does not meet these criteria.

The entire manuscript, including references, needs proof reading and correction for typographical errors, spelling, grammar, and consistency of tense and language.

Detailed comments

Abstract

- For the acquisition phase it would be useful to clarify how many “runs” dogs required during the one to four hours of training. Was it just a single presentation of each sample or were they presented multiple times during?

- Please clarify the wording around how many dogs were used for data collection in the study (it currently confusing for the reader whether 8 or 18 dogs were used for data collection).

- As per my general comments, I feel the conclusion that the sweat of SARS-CoV-2 contaminated persons can be detected by trained dogs is too strong at present, as the study does not appear to provide a robust double blind assessment and possible sources of bias/limitations have not been discussed. The study appears to be a good first step, but there are other possible explanations for the results based on the data presented. I suggest the conclusion is caveated that this study is a promising first step, and that a double blind trials is needed to more robustly confirm the results (unless this data is available and can be presented).

Medical detection dogs: state of play

- To avoid bias towards only discussing positive studies and to provide a more balanced discussion, I suggest mentioning some of the difficulties of conducting these studies where it is unknown whether the target odour exists, and that not all studies to date have shown disease odours to be reliable detectable by dogs. Issues are summarised in papers such as Elliker et al 2014, Edwards et al 2017, Johnen D, 2017.

Material and Methods

Samples

- Please clarify whether these 198 samples were each from individual people (i.e. 198 different people), or whether multiple samples were taken from the same person. Please could you clarify which samples were used for training and which were used for the trial stage (if training samples were reused for the trial stage this is a limitation that needs mentioning).

Practical realisation and storage of samples

- Please clarify if there is any risk that positive and negative samples could have been contaminated with different odours due to the way they were collected. For example, you state positive samples were collected with the person wearing full COVID-19 safety equipment – could this have transferred odour to the positive samples and not the negatives? Were negative samples also disinfected - if not, it is possible that dogs were training on the odour of disinfectant on the positives?

Canine Ressources (note spelling mistake in title)

The dogs

- The photos of dogs are nice but are not required in a scientific paper – suggest delete.

Training and first testing of the dogs

- Lines 262 to 263. The statement that the acquisition of the specific odour of COVIS-19+ sweat samples required from one to four hours ideally needs further supporting details on the training methods used and statistical evidence that the discrimination had been learnt (e.g. what was the learning criterion, were the handlers blind, were blank/negative samples also placed out, how many presentations did each dog require?). Please clarify if the learning criterion was based on the subjective opinion of the handlers rather than a statistical criterion.

- This seems to be extremely rapid acquisition (some medical detection dog studies have required weeks or months of training) so a comment on this would be useful in the discussion section.

- Were the training samples different to those used for actual tests? If not, please include as a study limitation as the dogs could have learnt the specific odours of the training samples rather than SARS-CoV-2 generally.

Testing protocol

- Lines 282-283. Repeated use of the same sample could be considered "pseudoreplication". Please clarify if this was taken into account during the data analysis (i.e. the 3 samples should not be considered as independent trials if the dog had 3 attempts to detect the same sample - it’s 3 attempts at a single "diagnosis"). Ideally each sample, positive or negative, should be used only once in the test. If they were used more than once this need to be discussed as an important study limitation, as the dogs may have learnt individual samples rather than generalising based on a SARS-CoV-2 odour.

- Lines 290-291. Please comment on whether there is any risk that the person putting out the samples could have unintentionally influenced the results by handling or storing the positive and negative samples differently. The dogs may learn minor differences in packaging odour, storage location or handling (e.g. positives and negatives stored in different fridges may carry different environmental odours that the dogs learn).

- The ideal experimental design would be to run the trial double blind (the person putting out the samples does not know if they are positive or negative during the trial). If it was not double blind this needs to be mentioned as a study limitation.

- Lines 292-293. Please comment on whether the individuals collecting data were blind to the sample allocations and whether they could have biased the results (ideally they should be blind too).

Lines 295-296. This sentence suggests that dogs had not previously been exposed to the samples used in the trials – please clarify if this is correct. Does this mean new samples were from new individuals not previously assessed by the dogs?

Results and statistical analysis

- Line 313. As above, if samples have been reused several times for the same dog this may be considered psuedoreplication of data. Please clarify if this was taken into account during the statistical analysis. Reuse of samples/individuals may result in detection based on learning of individual odours rather than COVID odour generalisation (it is not impossible that dogs could potentially learn 33 different individual samples)

Table 3: Results obtained on the dogs that finished the training period (n=8)

- Please state how many different positive samples (from different people) each dog was presented. It is not clear whether samples collected from the same individuals were reused more than once per dog. If they were reused, this needs to be included as a study limitation.

- Ideally it would be best to provide your underlying data (breakdown of trials for each dog showing positive and negative samples put out and response of the dog) as a supplementary file/download.

Discussion

- As per my general comments above, I recommend that the discussion is revised to highlight that this study a promising first step, but you include a more critical discussion of the possible limitations and possible alternative explanations for the dogs’ performance. I think the current conclusion (lines 331-332) is too strong based on the methods and data presented, without a more robust double-blind and independently validated trial taking place.

- Lines 359-382. Although interesting, I feel that the discussion of female hormone detection is moving too far away from the focus of the study subject and would be better deleted from the discussion (the evidence for the dogs indicating these samples due to female pheromones is also weak/anecdotal).

Conclusion

- As per my general comments above, I think the current conclusion (lines 417-419) is too strong based on the methods and data presented, without a more robust double-blind and independently validated trial taking place – suggest revision of wording in line with my general comments above.

Figures

- Figures 3. 4 and 11 are not readable due to poor resolution.

- Suggest delete figures 5 and 6.

6. PLOS authors have the option to publish the peer review history of their article (what does this mean?). If published, this will include your full peer review and any attached files.

Reviewer #1: No

Reviewer #2: No

Reviewer #3: No

---

## [Author Response · Author response to Decision Letter 0]

1 Sep 2020

Point-by-point response to reviewers.

Reviewer #1

This article investigates whether dogs can differentiate the axillary odor of individuals infected by SARS-CoV-2 from individuals that are not infected. The study is very well justified and potential applications for the management of the current pandemic (especially in case of a feared second wave) are highly relevant.

I however have some major issues with this study, as detailed below.

1) Design. 

With the current information provided in the method, the reader cannot be convinced that the authors controlled for a number of confounding factors. Namely, it should be made clear that the negative and positive samples cannot be different from each other based on other factors than infection to SARS-CoV-2. It is known that body odor changes according to medication intake, gender, age, health status, hormonal status etc. With this in mind, a number of questions can be raised here. Who are the COVID-negative patients: do they have other pathologies? If yes, which ones? Do they all have the same pathology, or different ones? Are they on medication? Are the COVID+ on medication and if yes, is it the same one for everybody? How are the negative samples chosen with regards to gender and age, according to the target (COVID+ sample) characteristics? Two odor sampling methods are mentioned (gauze swabs or filters and polymer tubes): Are they both used for negative and positive samples, and is the same sampling support used in the whole line during a trial? In sum, more information is expected in the method section on these aspects; the limits related to the possible confounding factors must be discussed in more details than it currently is; and the conclusion will probably need to be more cautious (as a consequence).

We thank the reviewer for this very relevant comment. 

The COVID-19 positive and negative samples were drawn from patients or individuals working in one of five hospitals (four located in the Paris or in the Suburb of Paris, France, and one located in Beirut, Lebanon), as now indicated in the manuscript. In each hospital, once a sample from a COVID-19 positive patient was collected (who fulfilled the following inclusion criteria: showing clinical symptoms of COVID-19, who were COVID-19 positive on RT-PCR or PCR test for SARS-CoV-2), a sample from a COVID-19 negative individual from the same hospital (patients or nursing staff) was collected (who fulfilled the following inclusion criteria: no clinical symptoms related to COVID-19 and negative on SARS-CoV-2 PCR test). Therefore, the potential confounding factor of the “hospital odor” was taken into account. We added this point in the Discussion section.

Regarding the comment about the odor sampling supports, first of all, for the training sessions as well as for the testing sessions, the same sampling support was used on each of the 3 or 4 cones of the lineup. Therefore, the dogs could not have been influenced by the odor of the sampling support when it indicated a cone. For one dog, only the polymer tubes were used both for the training and testing sessions. For the five other dogs, they were trained and tested on each of the two sampling supports. We added this important point raised by the reviewer in the Material and Method section, as well as in the Discussion section. 

Regarding the comparability criteria between COVID-19 positive and negative individuals, we added in the manuscript the percentage of female and the mean age between the two groups, for Paris and Beirut sites (the data from the Ajaccio site were excluded because there were not enough COVID-19 positive samples and these ones were used too many times for one dog). As the reviewer can see, the percentages of female are similar between the two groups (COVID-19 positives and COVID-19 negatives, both in Paris and Beirut). However, although similar between the two groups in Beirut site, the means of age were not similar in Paris site (48 years for the COVID-19 positive group versus 70 years for the COVID-19 negative group). We were able to collect data for some COVID-19 positive and negative individuals of the Parisian site on diabetes, overweight, arterial hypertension, respiratory disorders (such as asthma, or previous respiratory viral infections), and hypothyroidism. 

The results are the following:

 COVID-19 positive COVID-19 negative

Diabetes 4/14 (29%) 1/8 (13%)

Overweight 12/19 (63%) 4/11 (36%)

Arterial hypertension 9/17 (53%) 3/6 (50%)

Respiratory disorders 3/15 (20%) 3/4 (75%)

Hypothyroidism 4/14 (29%) 1/3 (33%)

Regarding the overweight, only one dog of the Parisian site (Gun) marked two (out of the four) COVID-19 negative but overweight individuals. The three other dogs did not mark these COVID-19 negative but overweight individuals.

Regarding the diabetes, the same dog (Gun) and only this dog (and not the other three) marked this COVID-19 negative individual having diabetes. 

Although these data are very scarce, these results do not provide strong evidence that dogs marked diabetes or overweight condition rather than COVID-19 condition. However, the low number of available data prevent us from providing such number in the manuscript and claiming such statement. We mentioned this point in the discussion section, and the fact that confounding bias cannot be ruled out.

2) Statistical analyses. 

The authors mention accuracy of the dog’s responses in introduction and conclusion, in terms of sensitivity and specificity. It should be clearly defined, especially with regards of the tests performed in the present article. The signal detection theory is evoked only in discussion, and false positives are discussed in that same section. It is therefore expected to see the current data analyzed using these criteria, which is not the case.

The word “accuracy” was mentioned in the Introduction section only when we referred to the paper by Maurer (Open Forum Infect Dis, 2016), inside which sensitivity and specificity terms were mentioned. This “accuracy” term has been removed from the manuscript to avoid any confusion by the readers. We now mentioned the sensitivity and specificity terms only when referring to subsequent studies. 

Our data collection does not allow us to calculate any sensitivity or specificity since a trial consisted of a lineup containing 3 or 4 cones, with only one positive sample on the lineup, and the dog had only one “guess” to mark it out of the 3 or 4 samples of the lineup. (We removed the “false positive” expression in the manuscript in order to avoid any confusion.) The success rate (ie, the number of correct indications divided by the total number of trials) was rather calculated with its 95% confidence interval. Again, this study is a proof-of-concept study, and it therefore presents some limitations compared to a confirmatory study in which sensitivities and specificities can be calculated. To make clear that the readers should not expect sensitivities and specificities, we indicated this point in the Material and Method section. 

3) Paragraph about human pheromones in the discussion (lines 359-382). 

I suggest to simply remove this paragraph, which is 1) erroneous (no pheromones have ever been identified in humans, and the androgenous compounds you are referring to are male – not female – compounds supposed to attract females), 2) very speculative (how did you estimate the fertile phase of the donor menstrual cycle?), 3) disproportionally long compared to other points to discuss that are more crucial and more directly related to the main question.

As also suggested by another reviewer, these four paragraphs were removed from the manuscript.

Minor points.

1) Please describe what the mocks are, and how many are there in the line.

We added the description of a mock in the Material and Methods section.

Please find the requested information below for each of the 4 dogs of the Parisian site.

Gun (47 trials, lineup of 3 cones): 44 trials with 0 mock and 3 trials with 1 mock.

Guess (52 trials, lineup of 4 cones): 34 trials with 1 mock and 18 trials with 2 mocks.

Maika (55 trials, lineup of 4 cones): 38 trials with 1 mock and 17 trials with 2 mocks.

Oslo (31 trials, lineup of 4 cones): 31 trials with 0 mock.

2) Please better explain how the training was conducted with regards to sample type: are there negatives at this stage? Mocks?

We agree that the training session was too scarcely described. We detailed it, and we hope that it is better explained.

3) Please explain body odor sample management. From what I understand, they are used fresh (minimum 24 hrs between odor collection and training). Are they thrown away after 3 trials? Are they frozen at any time in the process? Refrozen?

Any sample, whatever COVID-19 positive or negative, was never frozen. The samples were used fresh and thrown away after all dogs carried out the trials. The time between sample collection at the hospital and the trial (either during the training period or during the testing period) was independent of the sample status (COVID+ or COVID-).

Finally, on a more formal aspect, the article should be proofread for English. Also, improvement of the writing would be welcome. Some sentences are difficult to understand (i.e., lines 237-240 / 395-398) and some others do not integrate well in the chain of arguments (i.e. lines 62-64 / 343-345). Most importantly, in its current form the article is very fractionated (“bullet points” feeling in the whole document): It would very much gain in fluidity and articulation of the ideas if the authors work on this aspect.

We deeply revised the writing of the manuscript, and we do hope that its current form is now acceptable. 

 

Reviewer #2

The idea of using trained dogs for COVID-19 screening was boosted by mass media in many countries e.g. by the BBC in UK, and by announcing by many researchers and dog trainers, that they intend to train dogs to detect COVID-19. It is obvious that first published experimental works in this field, will have a great potential for being broadly cited or comercialized. However, it could be presumed that such experiments and manuscripts are probably accomplished under a strong pressure of time, to be published as the first. Therefore the papers like the present one, which, to my knowledge is the first paper on this topic, should be reviewed and evaluated particularly causciously and critically. It is important not to disseminate facts that are not based on sound scientific experiments. This is particularly important since the reviewed paper is already available online at https://www.biorxiv.org/content/10.1101/2020.06.03.132134v1.full.pdf as not certified by peer review.

For justification of their study the authors cited publications on cancer detecting dogs and dogs alerting hypoglycaemia and seisure in human patients. However, it should be mentioned that despite of about 20 publications within the last 17 years, the medical detection dogs have not been applied for practical clinical screening so far. This was evident on the latest international conference on cancer detection by dogs, organized by the Curie-Institute in Paris in October 2019. Also, despite of numerous chemical studies within more than 25 years, using gas chromatography / mass spectrometry, which found some differences in volatile organic compounds (VOCs) between cancer positive and cancer negative patients, no single VOC or a simple combination of VOCs were identified, that could be reliable cancer marker(s), mature for practical cancer screening. The reason for problematic application of methods based on identification of VOCs for disease screening, either by chemical analyses or by trained animals, is a great variability of emited VOCs due to day-to-day differences in diet, medication and numerous other confounding factors.

The aim of the reviewed study was to assess if dogs could be trained to discriminate body odor of patients infected with SARS-Cov-2, from body odor of healthy controls.

Such aim is justified by the current and probably future situation with COVID-19, untill an effective and commonly accessible COVID-19 vaccine is available.

The experiments were conducted at 3 different locations, with a total of 18 dogs which were previously trained either for explosives detection or for search and rescue or for cancer detection.

The Authors collected altogether 101 sweat samples from patients showing clinical symptoms of COVID-19, being also COVID-19 positive on RT-PCR or PCR test for SARS-CoV-2. Negative samples (n=97) were collected the same way, by the same trained staffs, from patients who have no clinical symptoms related to COVID-19, and were negative on SARS-CoV-2 PCR test.

The main flaw of collecting sweat samples for this study was, that the samples were not controlled quantitatively for the amount of the sweat. It is known that one of the typical COVID symptoms is a high fever. Humans with fever sweat more intensively than healthy people, so the amount of the sweat on the sample may be a clear cue for dogs.

As an example, our early studies in the eighties, on dogs detecting so called „silent estrus” in cows, could be mentioned. We used cotton swabs for collection vaginal mucus from cows in estrus vs diestrus. Our dogs perfectly indicated estrus samples. Only later we noticed that cows in estrus usually secreted more mucus and this was a cue for the dogs. When the dogs became more „experienced” they perfectly indicated cotton swabs moistured even with water. This made me causcious and critical about dog’s reliability. For dogs, odors like cow’s in estrus, human cancer, drugs or explosives, or SARS-CoV-2, play no biological role and such odors are indicated by dogs only because an operant conditioned response has been trained by producing association between odors and a reward in form a treat or favorite object to be retrieved and play with. Therefore, the trained dogs, especially when they are more „experienced”, will seek for a cue or a simple solution, to earn a reward. Such strategy of dogs may produce frequent false alerts. At identification of human scents by dogs it should be remembered that samples of human scent contain both individual component and other components as those related to diet, disease(s), medication, hygiene, cosmetics, background odors of the location as well as own odors of the sampling materials (odor carriers).

In lines 346-349 the authors report on two samples that were negative according to inclusion criteria for negative samples, but were indicated by two dogs as positive and the PCR re-test showed that in fact they were positive. This, however, is not a convincing evidence that the dogs sniffed out specific molecules induced by the SARS-CoV-2 virus presence. These two persons were not controlled for the amount of the sweat or body temperature measured precisely at the time of sample collection. Without knowing if the odor samples differ quantitatively in the amount of the sweat, it is not clear whether the dogs indicated a hypothetic, characteristic odor emited by COVID positive patients, or rather indicated samples from people with fever, who sweat more.

A recent study (Concha AR, Guest CM, Harris R, Pike TW, Feugier A, Zulch H, et al. Canine Olfactory Thresholds to Amyl Acetate in a Biomedical Detection Scenario. Frontiers in veterinary science. 2019;5:345) actually showed that the accuracy of detection can be very high even at low levels of concentrations. By analogy, we expect that small amounts of sweat in the sample would be as efficient (in dog’s detection) as large amounts of sweat. Furthermore, it is well known that large amounts of a substance that a dog must detect often provide poor results due to the saturation of its olfactory receptors. With the sampling procedure we used in our study, the average amount of sweat obtained was around 75 mg for a 20 minute-contact of a 2x2 inches gauze with the skin, and the same amount of sweat were obtained for cellulosic filters. The inter-individual variability in sweat quantities obtained after this 20 minute-contact were not related to the status of the sample (COVID-19 positive or negative), but rather on individual characteristics of the sampled persons. 

This important point raised by the reviewer was added in the Discussion section. 

The Authors mentioned that the odor samples were collected „at the same location”. This is methodologically sound but it should be precised what does it means „the same location” in this context ? Was it the same room or the same buiding or the same district ? The authors must be more precise here, because it is known that the dogs might be conditioned on the characteristic „hospital odors”, as shown by Walczak et al.(2012) in cancer detecting dogs, when the cancer positive samples were collected mostly in hospitals.

The COVID-19 positive and COVID-19 negative samples were collected from individuals from the same hospital (one COVID-19 positive sample was matched to one COVID-19 negative sample inside the same hospital). The COVID-19 negative samples were collected either from patients of the hospital or from the nursing staff of the hospital. We cannot rule out that inside the hospital, rooms of COVID-19 positive patients may have a different odor compared to rooms of COVID-19 negative patients. However, as Walczak et al. wrote in their 2012 paper, “hospital rooms may have a hospital odor, derived mainly from disinfectants, which may be a common component of all samples taken from donors who are inside a hospital.”. Therefore, in our data, there should not be confounding bias due the hospital odor. We added this point in the Material and Methods section, as well as in the Discussion section.

Another problem with COVID-positive sweat samples is that positive patients are staying either in infectious hospitals or in isolation wards that are probably intensively desinfected.

In addition, the access of healthy COVID-negative healthy donors (controls) to the same room where sick patients are staying, is hazardeous and strictly limited. This may be a serious constraint for collecting samples for further studies by other researchers.

The COVID-19 negative individuals never stayed for a long time in the same room as the COVID-19 positive patients (never for the COVID-19 negative patients, and during a short period of time for the COVID-19 negative nursing staff). Furthermore, as previously cited (Walczak et al. (2012)), the hospital disinfection protocol should be the same between rooms inside the hospital. 

The description of the training methods only in 3 lines is very scarce. Were the dogs rewarded in every training and testing trial and who and how gave a signal that the dog’s indication was correct?

We agree that the training protocol was scarcely described, and we described it now more in details. We also modified the description of the testing protocol, where now it is written that the dog was rewarded once it correctly marked a COVID-19 positive cone (the data recorder, who was in the same room, knew the result of the indication of the dog, and told the result to the dog handler right after the indication by the dog, allowing the dog handler to reward the dog if the dog correctly marked the cone). We added in the manuscript that the data recorder, although not in a blinded situation, was located into the room such that there was no visual contact between the data recorder and either the dog or the dog handler. 

Lines 219-220 - It is true that search and rescue dogs (SRD), trained to perform disaster and area search, mainly work through the scent of the sweat. However, it must be taken into consideration that SRD are trained for detection of a generalized human odor, without discriminating individual persons and persons’ specific odor e.g. odor(s) related to a disease. Such dogs are relatively easy to be re-trained to work in a scent lineup to discriminate odours of particular person, but a re-training, to be reliable, takes longer than 1-4 hours and 4-10 trials.

As written in our manuscript (but modified since the original submitted version), three types of detection dogs were used. (1) Three explosives detection dogs, which are trained to detect between 30 and 40 different types of explosives, and are used to work on a line of samples that they have to sniff individually. For such dogs, if COVID-19 positive samples have a specific odor, they only have to memorize only one additional odor in their “olfactory memory bank”. (2) One search and rescue dog, which is trained to perform disaster and area searches, and works through the scent of the sweat (Grandjean D., Haak R., Massey J., Pritchard C., Schuller P., Riviere S.; The search and recue dog, Aniwa ed., Paris, France, 2077; pp296). We must point out that such search and rescue dogs are trained for detection of a generalized human odor, without discriminating individual persons and persons’ specific odor (e.g. odor(s) related to a disease). However, such dogs can be easily re-trained to work in a scent lineup to discriminate odors of particular person, but a re-training, to be reliable, takes longer than for explosives detection dogs. (3) Two colon cancer detection dogs trained on rectal gases (Sarkis R, Khazen J, Issa M, Khazzaka A, Hilal G, Grandjean D. Dépistage du cancer colorectal par détection olfactive canine. J Chir Visc. 2017;154(Supp 1):22). These detection dogs are used to work with samples on lineups, and the acquisition of a new odor is fast.

In lines 262-264 the Authors stated: „…For each involved dog, the acquisition of the specific odour of COVIS-19+ sweat samples required from one to four hours, with an amount of positive samples sniffing ranging from four to ten….”

I am skeptical about the ability of each dog to learn precisely the task within such a short time. For example for cancer detecting dogs, the training period according to 13 papers published in the period 2003-2014, ranged from several weeks (Pickel et al. 2004) to 16 months (Cornu et al.2010), and on average took 8 months.

Even if the Authors could observe that their dogs correctly indicated the target samples after one hour, performing only four trials from being confronted for the first time with COVID sweat samples, it seems inredible that dogs could acquire the ability to detect COVID odor so quickly. It can not be excluded that the dogs indicated samples in the lineup, that are merely different from the others e,g, by containing more sweat.

A recent study on COVID-19 detection dogs published by Jendrny et al. (Jendrny P, Schulz C, Twele F, Meller S, von Kockritz-Blickwede M, Osterhaus A, et al. Scent dog identification of samples from COVID-19 patients - a pilot study. BMC infectious diseases. 2020;20(1):536.) mentioned a similar time to train COVID-19 detection dogs. Of course, several weeks (depending on dog's ability and behavior) are necessary but are enough (between one and three for our six dogs). From the experience we now have with our collaborators of 5 different countries (Emirates, Brazil, Argentina, Chile, Mexico), the training session for one dog already trained for olfactory detection takes also between one and three weeks, and take between four and eight weeks for working dogs without any detection experience. 

Positive samples do not contain more sweat than negative. We saw that in Lebanese as well as in Emirians studies, where negative individuals do sweat as much (and sometimes even more) than COVID-19 positive hospitalized ones. 

In lines 265-267 or in lines 294-296 the Authors did not precise what was a criterion (percentage of correct indications, false alerts, misses, regularity of sniffing the samples in the lineup ?) for considering the dog to be fully trained and ready for testing.

In the more detailed training protocol paragraph, we added the requested information: “The dog was considered as trained and ready for the testing session by the subjective opinion of the handler, based on the behavior of the dog”.

From the lines 294-296 it could be concluded that separate samples were used for the training and testing, which is methodically correct. However, there is no information how many positive and negative or mock samples, out of the totals given in Table 1, were used for the training and real testing.

We decided to remove Table 1 and to replace it by the text in the Results section describing the study sample of COVID-19 negative and positive individuals, both for the Parisian site and for the Beirut site. If such Table of individuals would have still been present in the manuscript, it would have been uneasy to read if we added data regarding the trials (how many positive and negative or mock samples were used per dog). However, we described more in details the training and testing protocols. Furthermore, here are the data requested by the reviewer for the 4 Parisian dogs:

Gun (47 trials, lineup of 3 cones): 44 trials with 0 mock and 3 trials with 1 mock.

Out of the 5 unsuccessful trials, Guess marked a COVID-19 negative sample 5 times.

Guess (52 trials, lineup of 4 cones): 34 trials with 1 mock and 18 trials with 2 mocks.

Out of the 9 unsuccessful trials, Guess marked the mock 3 times, and a COVID-19 negative sample 6 times.

Maika (55 trials, lineup of 4 cones): 38 trials with 1 mock and 17 trials with 2 mocks.

Out of the 10 unsuccessful trials, Maika marked the mock 3 times, and a COVID-19 negative sample 7 times.

Oslo (31 trials, lineup of 4 cones): 31 trials with 0 mock.

We do hope that this addresses the issue raised by the reviewer. 

The authors did not conduct so called „zero trial” with no target samples in the lineup. This is another flaw which could have a consequence by making the overall conclusion in lines 407-408 premature. Although negative and mock samples were used in the lineup (lines 269-270), but there is no explanation what kinds of mocks were used. Were they samples with more sweat ? or samples from donors with fever but not related to COVID-19 (for example from people with seasonal influenza) ? „Zero trials” usually provoke more false alerts, especially on samples that are simple different from the others.

The lineup contained one (and only one) COVID-19 positive sample. All the other samples were negative samples or mock(s). The mock contained only the sample support (ie, the sterile gauze or the inert polymers tube). We added this information in the paragraph detailing the training protocol. As written in the manuscript, the inclusion criteria for COVID-19 negative individuals were the following: no clinical symptoms related to COVID-19, and having a negative PCR test for SARS-CoV-2. COVID-19 negative individuals may therefore not have fever. The absence of “zero trial” may prevent us from drawing the written conclusion (in lines 407-408 in the original version of the manuscript). We modified such conclusion accordingly. 

Minor remarks:

The manuscript should be better structured. For example the major portion of the Conclusion is appropriate rather for the Discussion section, whereas the real conclusion is given as the last sentence or the Discussion (in lines 407-408)

We modified the structure of the manuscript, and especially the Discussion section. We removed the Conclusion section. 

Line 64 More original papers should be cited instead of a handbook.

The paragraph where the handbook was cited has been removed.

Lines 76-78 „…Scent detection by animals has been published in a large number of diagnostic studies, which all suggest similar or even superior accuracy compared with standard diagnostic methods…” – nevertheless, the scent detection by animals is not used for medical diagnoses or screenings on a large scale (probably except for tuberculosis detecting Gambian pouched rats in Africa). The medical diagnosis of a disease should not be confused with screening. Scent detection by animals will never be a diagnosis.

We agree with the reviewer and modified this sentence accordingly.

Line 83 „…The first clinical investigation of cancer was published by Willis on bladder cancer in 2010, after having published a proof of principle study in 2004…” – this information is only partly true: McCulloch et al. published their study on canine detection of lung and breast cancer in 2006.

Actually, the sentence in the manuscript was about bladder cancer, not all cancers. However, we acknowledge that the sentence was not clear, and we modified it by adding the study of McCulloch suggested by the reviewer.

Lines 88-89 „….The dog’s nose is also now currently utilised in order to prevent the consequences of crisis (or even prevent them) for diabetic [30, 31, 32, 33] and epileptic people [70]….” - The accuracy and reliability of hypoglycaemia and seizure alerting dogs, when considering on a daily basis, is still questionable. The alerting dogs rather contribute to feeling more safe and thus to improving the quality of life of the patients, than to precise alerting the hypoglycaemia or seizure episodes.

We modified the sentence according to the comment of the reviewer.

Lines 115-117 „….In a second time……” – this assumption is not quite clearly formulated in English.

We modified the sentence.

Lines 154-158 Why two types of sampling supports are used ?:

(1) - Sterile gauze swabs utilised by the hospitals or sterile gauze filters utilised for explosive detection, and (2)- Inert polymers tubes utilised for explosives, drugs or criminology detection)

First of all, and most importantly, for the training period as well as for the testing period, the same sampling support was used on each of the 3 or 4 cones of the lineup. Therefore, the dogs could not have been influenced by the odor of the sampling support when it indicated a cone. For one dog, only the polymer tubes were used both for the training period as well as for the testing period. For the five other dogs, they were trained and carried out the testing session on each of the two sampling supports. Two sampling supports were used because at the time to the beginning of the proof-of-concept study, we did not know which one of the two sampling supports was the most effective. However, the main point is that the same sampling support was used on each of the 3 or 4 cones of the lineup.

We added this point raised by the reviewer in the Material and Method section, as well as in the Discussion section. 

Line 289 Why acetone was used for disinfection of sample carriers ? Acetone is an irritant, causing mild skin irritation and moderate to severe eye irritation which may be repelling for dogs.

We modified the sentence since it originally did not well describe the used procedure. It is now: “All cones were cleaned with straight water after each trial, were twice a day cleaned and disinfected with 10p100 aqueous acetone solution, and then dried up.”

Lines 292-297 „…A failure was defined if the dog alerts on any of the other boxes. Trials were registered from the moment dogs recognized positive sample among other samples (mock or negative ones) without having sniffed any of them previously to the test. If the dog marks first a negative sample, the trial is considered as failed, even if he marks the positive afterwards….”

- the authors did not precise whether the dogs have to sniff ALL samples prior to the indication. Have the dogs to sniff the remaining samples after the correct indication? If so, would the trial be declared as a failure, if the dog made a false alert AFTER prior correct indication in the same trial? This makes that the calculation of correct indication by chance should take into account how many samples a dog actually has sniffed. Therefore, when the dog indicates correctly for example on the first sample in the lineup and does not sniff the remaining samples, the probability of correct indication by chance at 7 samples in a lineup is not 14% but 50%. This would mean that the probability of correct indication by chance is varying from trial to trial?

First of all, the dog was trained such that it knew that there was only one positive sample on the lineup. Once the dog was trained and ready for the testing period, it was presented into the room where the samples were already and randomly placed behind each cone. Only one cone out of the cones contained the positive sample. All the other cones contained negative samples or mock. The dog was first presented to the first cone (the one on the very left of the room), then to the second (the next one on the right), and so on. The dog could freely sniff all the cones, but the rule was the following: once the dog indicates a cone (by sitting in front of it), the trial stops. If the dog indicates a cone with the positive sample, the trial was considered as a success; it was considered as a failure in the other case. In such a protocol, the calculation of the probability of correct indication by chance is still one divided by the number of cones on the lineup. For instance, in a lineup of four cones, if the first cone (ie, the one on the very left into the room) contained the positive sample, and if the dog indicates this first cone, the probability of indicating this cone by chance only is still 25%, because the trial consists in only one sit by the dog (again, the trail stops once the dog sits). The game for the dog was: “pick the only one positive sample out of the 3 or 4 samples on the lineup, and you have only one trial”. However, we acknowledge that the description of the protocol was not clear enough, and we modified it for better understanding of the protocol. We do hope that it is clearer now.

Line 350 „….dogs have no reason to lie….” – this is too colloquially expressed. With regards to odor detecting dogs one should not speak about "lying" or "telling truth".

>From the dog's perspective, detecting trained odors as for example the hypothetical COVID-19 odor, that have no biological relevance, is a rather a game to earn a reward. Dogs, even if they are well trained, when they are getting more routined, will try to get a reward at a simple way, which means that they will try to find any additional cues to indicate the target odor in the lineup. Therefore the dogs' performance in the scent lineup has to be permanently controlled throughout the whole deployment period, because it may vary. This seems to be one of the reason why so called medical detection dogs have not been used in medical screening on a regular basis.

We agree with the reviewer and we modified the expression “dogs have no reason to lie”. 

Table 3. The terms detection sensitivity and specificity, which derive from the point 1-4 in lines 385-388, should be given, or percentage of false alerts should be given instead of the column with the number of correct indications, which is redundant in this table

Neither Fjellanger et al. (ref 65) nor Concha et al. (ref 67) mentioned the sensitivity (Se) and specificity (Sp) terms in their study – we removed ref 66 in our manuscript, only because it was a handbook and not an original paper. In our study, we deliberately did not mention these terms of Se and Sp, simply because Se and Sp could not have been calculated with such our protocol. In order to calculate Se and Sp, we would have needed one (positive or negative) sample behind one cone only, and the dog would have had to sit, or not, in front of this specific cone (exactly like a diagnostic tool evaluating the presence of a disease from one sample of one patient). In our study, because the lineup contained 3 or 4 cones, all of them minus one containing negative samples or mocks, it was not possible to calculate such Se and Sp. In the same way, it is not a question of “false alert” (again, it would have been the case if there had been only one cone, either positive or negative). This study is a proof of concept study. It has not the strength of a study in which Se and Sp (or percentage of false alerts) can be calculated. Calculating such statistical indicators from our data would have led to an over-interpretation of our results. However, the 95% confidence intervals of the success rates we were able to calculate never included the success rate that would have been observed by chance, which provides some evidence that dogs seem a promising tool to detect COVID-19 positive people. We added this point in the Discussion section.

The English of the manuscript is for me as a non-native speaker understandable but suffers from many language and wording imperfections. It should be edited by a native English speaker, who also has some knowledge in the specific study field.

We modified the writing of the manuscript to avoid wording imperfections.

Line 48 „..State of play..” ? – rather „State-of-the-art.”

We removed this expression.

Line 51 What does it means „..for those of interest in dog’s olfaction…” in this context?

We modified the sentence.

Line 148. „…In order to avoid any dog creancement (?) on the olfactive “background noise

In several places of the manuscript there are statements that are difficult to follow.

We modified the sentence.

In lines 115-117 the authors write: „…In a second time, we attempt to validate the method with the dogs checking samples one by one instead of comparatively, in order to fit with what could be an operational utilisation of such trained dogs...” – what does it means exactly?

We removed this sentence because it finally appeared that is was not relevant.

Line 372 „ In Sheakspeare’s day…” ? - rather is „ in Sheakspeare’s era” or „in Sheakspeare’s time”

Lines 372-374 the passage with the anecdote on peeled apple under woman’s arm should be deleted because it is too colloquial and not closely related to the reviewed study.

As suggested by another reviewer, we removed the four paragraph regarding pheromones, including the one where the two expression cited above were written.

Line 404 „ …cynotechnicians…” ? or dog handlers ?

We modified the word “cynotechnicians” by dog handlers.

Line 455 „…by small…” ? – should be „ by smell”

The reviewer was right. We however removed the paragraph including the reference of the book since it was redundant with the following one in the manuscript.

In conclusion, this first report on dogs trained to detect COVID-19 is interesting but a flaw in the methodology of collecting scent samples (no control for the amount of the sweat) makes that the conclusion that „…there is a high evidence that dogs can detect a person infected by the virus responsible for COVID-19 disease …” seems to be premature.

We agree with the reviewer that the conclusion was too strong and premature regarding the protocol of the proof-of-concept study. We modified the discussion section accordingly.

In addition, some language imperfections make me to evaluate the manuscript as not acceptable for publication in PloS ONE in the present form and needing a major revision before resubmission.

First of all, in the major revision the issue of the amount of sweat in odor samples should be addressed, including replication of the trials.

Regarding the replication of the trials, we are conducting a second study which confirms the results of this proof-of-concept study. However, we needed to conduct and write the manuscript of this proof-of-concept study first, in order to implement the second one.

Regarding the issue about the sweat quantity, we answered the reviewer that, first, if the quantity of sweat has any impact of dog’s detection, it would be on the opposite way as the one suggested by the reviewer (ie, larger amounts of sweat would reduce the ability of the dog to detect a COVID-19 positive sample), and second, that the quantity of sweat was not related to the status of the sample but rather to the individual characteristics. We do hope that our explanations answered the issues raised by the reviewer. 

We added in the Material and Method section that some samples could have been used in different trials for one dog. This is an important point we discussed in the Discussion section.

 

Reviewer #3

Thank you for conducting and reporting this very interesting study. In my opinion, this study is a good first step towards testing whether dogs can detect odours associated with SARS-CoV-2 infection. You have used a reasonable sample size of dogs and SARS-CoV-2 samples and the methods are generally appropriate. A main difficulty with any study of this type, where it is not known whether the odour the dogs are being trained to detect exists (in this case SARS-CoV-2 related), is demonstrating beyond reasonable doubt that dogs are detecting the odour rather than incidental contamination, and demonstrating robustly that the research team is not unintentionally influencing/cuing the dogs. Sources of odour contamination may include, for example, medication used in SARS-CoV-2 patients, hospital odours, differences in the odour of the storage containers of positive and negative samples and different odours arising from how positive and negative samples are handled. You have taken some steps to try to control for these issues. However, in my opinion, it is not possible to conclusively say that the sweat of SARS-CoV-2 contaminated persons can be detected by trained dogs based on the methods and results you present so far. In places, the methodology and controls you used just need further clarification in order to rule out possible sources of bias and alternative explanations that the dogs may be detecting contaminating odours associated with the SARS-CoV-2 samples.

Most importantly, to support the conclusion that the sweat of SARS-CoV-2 contaminated persons can be detected by trained dogs, a more rigourous, double-blind test is required, in which the research team is provided with new samples from an independent source which they are asked to assess using the dogs. The dog handlers, sample handlers and data recorders (everyone involved in assessing the samples) should be blind to the sample identities (positive or negative) until the dogs have conducted the test. Methods for how to do this have been presented in a number of previous medical detection dog studies.

I would suggest two options for revising the paper:

- Additional data is added, providing the results of a double-blind test using samples from new people that have not previously been screened by the dogs, or

- The current abstract, discussion and conclusions are revised to provide a more critical appraisal of the results, covering possible limitations of the study and whether there are any other possible explanations for the dogs' detection performance and possible sources of unintentional bias.

Potential sources of bias/limitations in medical detection dogs studies have been summarised previously (e.g. Elliker et al 2014. Key considerations for the experimental training and evaluation of cancer odour detection dogs; Edwards et al 2017 Animal olfactory detection of human diseases: Guidelines and systematic review; Johnen D, 2017. An approach to identify bias in scent detection dog testing). I recommend you review these guidelines and report where your study meets these criteria, and/or discuss limitations where your study does not meet these criteria.

We thank a lot the reviewer for her/his very relevant comments. 

We chose the second option the reviewer suggested. At the mean time, we deeply modified the manuscript to clarify some points raised by the reviewer. The potential confounding bias due to “hospital odor” was already taken into account thanks to the matching of COVID-19 positive individuals to COVID-19 negative individuals from the same hospital (this important information has been added in the manuscript). 

We also added in the manuscript that both the dog handler and the sample handler were blinded to the sample identity (COVID-19 positive or negative). The data recorder was not blinded but was located in the back of the room, where neither the dog handler nor the dog could see the data recorder when the dog was searching the positive sample. 

Regarding the comparability criteria between COVID-19 positive and negative individuals, we added in the manuscript the percentage of female and the mean age between the two groups, for Paris and Beirut sites (the data from the Ajaccio site were excluded because there were not enough COVID-19 positive samples and these ones were used too many times for one dog). As the reviewer can see, the percentages of female are similar between the two groups, both in Paris and Beirut. However, although similar between the two groups in Beirut site, the means of age were not similar in Paris site (48 years for the COVID-19 positive group versus 70 years for the COVID-19 negative group). We were able to collect data for some COVID-19 positive and negative individuals of the Parisian site on diabetes, overweight, arterial hypertension, respiratory disorders (such as asthma, or previous respiratory viral infections), and hypothyroidism. 

The results are the following:

 COVID-19 positive COVID-19 negative

Diabetes 4/14 (29%) 1/8 (13%)

Overweight 12/19 (63%) 4/11 (36%)

Arterial hypertension 9/17 (53%) 3/6 (50%)

Respiratory disorders 3/15 (20%) 3/4 (75%)

Hypothyroidism 4/14 (29%) 1/3 (33%)

Regarding the overweight, only one dog of the Parisian site (Gun) marked two (out of the four) COVID-19 negative but overweight individuals. The three other dogs did not mark these COVID-19 negative but overweight individuals.

Regarding the diabetes, the same dog (Gun) and only this dog (and not the other three) marked this COVID-19 negative individual having diabetes. 

Although these data are very scarce, these results are not in favor of that dogs mark diabetes or overweight condition rather than COVID-19 condition. However, the low number of available data prevent us from providing such number in the manuscript. We will mention this point in the discussion section, and the fact that, as the reviewer wrote, confounding bias cannot be ruled out.

We also thank the reviewer for the suggested references, which are now cited in the manuscript. 

The entire manuscript, including references, needs proof reading and correction for typographical errors, spelling, grammar, and consistency of tense and language.

We modified the writing of the manuscript and we hope that it is now suitable for the reviewer.

Detailed comments

Abstract

- For the acquisition phase it would be useful to clarify how many “runs” dogs required during the one to four hours of training. Was it just a single presentation of each sample or were they presented multiple times during?

The training protocol is now described more in details in the Material and Methods section.

- Please clarify the wording around how many dogs were used for data collection in the study (it currently confusing for the reader whether 8 or 18 dogs were used for data collection).

After the removal of the data from the Ajaccio site, 14 dogs were trained and 6 dogs finally performed the trials for testing. We only mention now the six dogs used for the study; we did not mention anymore the number of dogs used for the training, since it added a non relevant information. The new Table 1 (ex Table 2) contained only the six dogs which performed the testing session. 

- As per my general comments, I feel the conclusion that the sweat of SARS-CoV-2 contaminated persons can be detected by trained dogs is too strong at present, as the study does not appear to provide a robust double blind assessment and possible sources of bias/limitations have not been discussed. The study appears to be a good first step, but there are other possible explanations for the results based on the data presented. I suggest the conclusion is caveated that this study is a promising first step, and that a double blind trials is needed to more robustly confirm the results (unless this data is available and can be presented).

We totally agree with the reviewer, and we modified the Discussion section and the conclusion sentence accordingly. 

Medical detection dogs: state of play

- To avoid bias towards only discussing positive studies and to provide a more balanced discussion, I suggest mentioning some of the difficulties of conducting these studies where it is unknown whether the target odour exists, and that not all studies to date have shown disease odours to be reliable detectable by dogs. Issues are summarised in papers such as Elliker et al 2014, Edwards et al 2017, Johnen D, 2017.

We thank the reviewer and added the points raised in the three mentioned papers in the manuscript.

Material and Methods

Samples

- Please clarify whether these 198 samples were each from individual people (i.e. 198 different people), or whether multiple samples were taken from the same person. 

There were 177 samples collected from 177 different individuals (one sample per individual). We added the clarification into the manuscript.

Please could you clarify which samples were used for training and which were used for the trial stage (if training samples were reused for the trial stage this is a limitation that needs mentioning).

Actually, samples used for the training sessions were not used the testing sessions. This point has been clarified at the beginning of the testing protocol paragraph. 

Practical realisation and storage of samples

- Please clarify if there is any risk that positive and negative samples could have been contaminated with different odours due to the way they were collected. For example, you state positive samples were collected with the person wearing full COVID-19 safety equipment – could this have transferred odour to the positive samples and not the negatives? Were negative samples also disinfected - if not, it is possible that dogs were training on the odour of disinfectant on the positives?

There was no risk of odors contamination between positive and negative samples. Worn equipment was the same for positives and negatives. However, we agree that the original text was not clear enough about this subject, and modified it accordingly (see “Samples” part of the manuscript).

Canine Ressources (note spelling mistake in title)

The dogs

- The photos of dogs are nice but are not required in a scientific paper – suggest delete.

We deleted the photos.

Training and first testing of the dogs

- Lines 262 to 263. The statement that the acquisition of the specific odour of COVIS-19+ sweat samples required from one to four hours ideally needs further supporting details on the training methods used and statistical evidence that the discrimination had been learnt (e.g. what was the learning criterion, were the handlers blind, were blank/negative samples also placed out, how many presentations did each dog require?). Please clarify if the learning criterion was based on the subjective opinion of the handlers rather than a statistical criterion.

We developed this training protocol of the manuscript, in order to provide more information for the reader. Regarding the learning criterion, we added: “The dog was considered as trained and ready for the testing session by the subjective opinion of the handler, based on the behavior of the dog.”

- This seems to be extremely rapid acquisition (some medical detection dog studies have required weeks or months of training) so a comment on this would be useful in the discussion section.

As written in our manuscript (but modified since the original submitted version), three types of detection dogs were used. (1) Three explosives detection dogs, which are trained to detect between 30 and 40 different types of explosives, and are used to work on a line of samples that they have to sniff individually. For such dogs, if COVID-19 positive samples have a specific odor, they only have to memorize only one additional odor in their “olfactory memory bank”. (2) One search and rescue dog, which is trained to perform disaster and area searches, and works through the scent of the sweat (Grandjean D., Haak R., Massey J., Pritchard C., Schuller P., Riviere S.; The search and recue dog, Aniwa ed., Paris, France, 2077; pp296). We must point out that such search and rescue dogs are trained for detection of a generalized human odor, without discriminating individual persons and persons’ specific odor (e.g. odor(s) related to a disease). However, such dogs can be easily re-trained to work in a scent lineup to discriminate odors of particular person, but a re-training, to be reliable, takes longer than for explosives detection dogs. (3) Two colon cancer detection dogs trained on rectal gases (Sarkis R, Khazen J, Issa M, Khazzaka A, Hilal G, Grandjean D. Dépistage du cancer colorectal par détection olfactive canine. J Chir Visc. 2017;154(Supp 1):22). These detection dogs are used to work with samples on lineups, and the acquisition of a new odor is fast.

From the experience we now have with our collaborators of 5 different countries (Emirates, Brazil, Argentina, Chile, Mexico), the training session for one dog already trained for olfactory detection takes also between one and three weeks, and take between four and eight weeks for working dogs without any detection experience.

A recent study published by Jendrny et al. on COVID-19 detection dogs (Jendrny P, Schulz C, Twele F, Meller S, von Kockritz-Blickwede M, Osterhaus A, et al. Scent dog identification of samples from COVID-19 patients - a pilot study. BMC infectious diseases. 2020;20(1):536.) mentioned a similar time to train COVID-19 detection dogs. 

We added this point in the Discussion section. 

- Were the training samples different to those used for actual tests? If not, please include as a study limitation as the dogs could have learnt the specific odours of the training samples rather than SARS-CoV-2 generally.

As previously mentioned, samples for training were not used for testing sessions. 

Testing protocol

- Lines 282-283. Repeated use of the same sample could be considered "pseudoreplication". Please clarify if this was taken into account during the data analysis (i.e. the 3 samples should not be considered as independent trials if the dog had 3 attempts to detect the same sample - it’s 3 attempts at a single "diagnosis"). Ideally each sample, positive or negative, should be used only once in the test. If they were used more than once this need to be discussed as an important study limitation, as the dogs may have learnt individual samples rather than generalising based on a SARS-CoV-2 odour.

We thank the reviewer for the relevant comment. COVID-19 positive or negative samples could have been used up to three times for one dog, during the testing session. As the reviewer said, this is a limitation of the study that we discuss now the in Discussion section. Importantly, because Williams et al. showed that the dogs can learn at least 10 odors (Williams M, Johnston JM. Training and maintaining the performance of dogs (Canis familiaris) on an increasing number of odor discriminations in a controlled setting. Appl Anim Behav Sci. 2002;78(1):55-65), we removed the data from the Ajaccio site since there were only 6 COVID-19 positive samples (and 15 COVID-19 negative samples). In such a case, we cannot rule out, based on the study of Williams et al., that the dogs from Ajaccio used their olfactory memory to detect COVID-19 positive samples. However, for the remaining sites (Paris and Beirut), the dogs worked on at least 24 COVID-19 positive samples (and at least 33 COVID-19 negative samples), and we make the hypothesis according to which the olfactory memory of dog did not play a major role in the detection task, but may have been able to generalize the odor of COVID-19 positive samples used for training. We added this important point in the paragraph describing the testing session as well as in the Discussion section.

- Lines 290-291. Please comment on whether there is any risk that the person putting out the samples could have unintentionally influenced the results by handling or storing the positive and negative samples differently. The dogs may learn minor differences in packaging odour, storage location or handling (e.g. positives and negatives stored in different fridges may carry different environmental odours that the dogs learn).

Positive and negative samples were collected in the same way by the person. They were also stored in the same fridges. Therefore, the positive and negative samples coming from the same hospital (they were matched on hospital) were in the same environment from collection to sniffing by the dog. We clarified this point in the Material and Methods section.

- The ideal experimental design would be to run the trial double blind (the person putting out the samples does not know if they are positive or negative during the trial). If it was not double blind this needs to be mentioned as a study limitation.

Both the dog handler as well as the sample handler were blinded to the sample status (COVID-19 positive or negative). We added this point in the manuscript, in the testing protocol paragraph.

- Lines 292-293. Please comment on whether the individuals collecting data were blind to the sample allocations and whether they could have biased the results (ideally they should be blind too).

The data recorder was not blinded to the sample identity. However, he was located in the back of the room, where neither the dog handler nor the dog could see the data recorder when the dog was searching the positive sample. According to us, he could not have influenced the behavior of the dog. Furthermore, this absence of blindness allowed the data recorder to tell the handler whether the trial was successful or not, allowing to reward the dog each time it corrected marked the cone. This clarification was added in the testing protocol paragraph. We added this point in the Discussion section.

Lines 295-296. This sentence suggests that dogs had not previously been exposed to the samples used in the trials – please clarify if this is correct. Does this mean new samples were from new individuals not previously assessed by the dogs?

Yes, this is correct. We clarified this point in the testing protocol paragraph. 

Results and statistical analysis

- Line 313. As above, if samples have been reused several times for the same dog this may be considered psuedoreplication of data. Please clarify if this was taken into account during the statistical analysis. Reuse of samples/individuals may result in detection based on learning of individual odours rather than COVID odour generalisation (it is not impossible that dogs could potentially learn 33 different individual samples)

We do agree that because one sample could have been used up to three times, it is not impossible, as the reviewer said, that the dogs used their olfactory memory to detect COVID-19 positive samples. However, we checked in the Parisian data the number of trials between two trials using the same COVID-19 positive sample. Here are the results for the 4 dogs regarding the COVID-positive samples:

Guess: 4 COVID+ samples were used once, 21 were used twice, and 2 were used 3 times.

Median number between two trials using the same COVID+ sample [p25-p75] (min-max): 7 [5-10] (3-22)

Maika: 3 COVID+ samples were used once, 20 were used twice, and 4 were used 3 times.

Median number between two trials using the same COVID+ sample [p25-p75] (min-max): 6 [4-10] (2-27)

Gun: 5 COVID+ samples were used once, 21 were used twice.

Median number between two trials using the same COVID+ sample [p25-p75] (min-max): 8 [4-12] (2-21)

Oslo: 11 COVID+ samples were used once, 10 were used twice.

Median number between two trials using the same COVID+ sample [p25-p75] (min-max): 3 [1-4] (1-9)

For three out of the four dogs (Guess, Maika, and Gun), the “time” interval between two identical COVID-19 positive samples seems enough to make the hypothesis that the olfactory memory played in minor role in the detection positive cones. For the last dog (Oslo), the high number of COVID+ samples used only once counterbalances the fact that the 1st quartile of the number between two trials using the same COVID+ sample is equal to 1.

Although we did not include these numbers in the manuscript, we added this important point about the repeated use of samples in the Discussion section.

Table 3: Results obtained on the dogs that finished the training period (n=8)

- Please state how many different positive samples (from different people) each dog was presented. It is not clear whether samples collected from the same individuals were reused more than once per dog. If they were reused, this needs to be included as a study limitation.

We indicate now the number of COVID-19 positive and negative individuals which were used for the study in the Results section (n=177). We also added in the testing protocol paragraph that one sample could have been used in up to three trials for one dog. We added this point in the Discussion section.

- Ideally it would be best to provide your underlying data (breakdown of trials for each dog showing positive and negative samples put out and response of the dog) as a supplementary file/download.

The data for Paris site are provided to the Editor. Because of the recent exploding event in Beirut, it was not possible to obtain the data from the testing sessions in Beirut. 

Discussion 

- As per my general comments above, I recommend that the discussion is revised to highlight that this study a promising first step, but you include a more critical discussion of the possible limitations and possible alternative explanations for the dogs’ performance. I think the current conclusion (lines 331-332) is too strong based on the methods and data presented, without a more robust double-blind and independently validated trial taking place.

We agree with the reviewer, and modified the Discussion section accordingly.

- Lines 359-382. Although interesting, I feel that the discussion of female hormone detection is moving too far away from the focus of the study subject and would be better deleted from the discussion (the evidence for the dogs indicating these samples due to female pheromones is also weak/anecdotal).

As suggested also by another reviewer, we deleted the four paragraphs regarding female hormone detection.

Conclusion

- As per my general comments above, I think the current conclusion (lines 417-419) is too strong based on the methods and data presented, without a more robust double-blind and independently validated trial taking place – suggest revision of wording in line with my general comments above.

We modified the Discussion section and the conclusion sentence.

Figures

- Figures 3. 4 and 11 are not readable due to poor resolution.

We improved the resolution of the figures mentioned above.

- Suggest delete figures 5 and 6.

We deleted these two figures.

---

## [Decision Letter · Decision Letter 1]

29 Sep 2020

PONE-D-20-17256R1

Can the detection dog alert on COVID-19 positive persons by sniffing axillary sweat samples?

A proof-of concept study

PLOS ONE

Dear Dr. grandjean,

Thank you for submitting your manuscript to PLOS ONE. After careful consideration, we feel that it has merit but does not fully meet PLOS ONE’s publication criteria as it currently stands. Therefore, we invite you to submit a revised version of the manuscript that addresses the points raised during the review process.

Two reviewers ar still requiring some clarification or some revision of your conclusions and importantly recognize some indications of the limits of your study.

We look forward to receiving your revised manuscript.

Kind regards,

Nadine Ravel

Academic Editor

PLOS ONE

Reviewers' comments:

Reviewer's Responses to Questions

**Comments to the Author**

1. If the authors have adequately addressed your comments raised in a previous round of review and you feel that this manuscript is now acceptable for publication, you may indicate that here to bypass the “Comments to the Author” section, enter your conflict of interest statement in the “Confidential to Editor” section, and submit your "Accept" recommendation.

Reviewer #1: (No Response)

Reviewer #2: All comments have been addressed

Reviewer #3: (No Response)

2. Is the manuscript technically sound, and do the data support the conclusions?

Reviewer #1: Partly

Reviewer #2: Yes

Reviewer #3: Partly

3. Has the statistical analysis been performed appropriately and rigorously? 

Reviewer #1: I Don't Know

Reviewer #2: Yes

Reviewer #3: No

4. Have the authors made all data underlying the findings in their manuscript fully available?

Reviewer #1: No

Reviewer #2: Yes

Reviewer #3: Yes

5. Is the manuscript presented in an intelligible fashion and written in standard English?

Reviewer #1: No

Reviewer #2: Yes

Reviewer #3: Yes

6. Review Comments to the Author

Reviewer #1: The authors have considerably improved the manuscript, by providing significant information that was lacking in the previous version, and by improving the structure of the manuscript. For this kind of study, and especially due to the potential impact its results may have on the current management of the Covid-19 disease, it remains essential that the methodology is carefully discussed, so that the validity of the conclusions can be properly evaluated. In the current version, the additional details provided have elicited further questions about the validity of the conclusions:

1) The pairing between COVID-19 positive and COVID-19 negative odor donors represents a strong limitation in this study. Whereas the Lebanese groups are comparable, age differences between both groups in Paris is considerable: as body odor differs with age (to human noses as well as in terms of chemical composition: Mitro et al 2012 “The smell of age: perception and discrimination of body odors of different ages”, Chen & Haviland-Jones 2000 “Human olfactory communication of emotion”, Haze et al 2001 “2-Nonenal newly found in human body odor tends to increase with aging”), this may interfere with the dogs’ discrimination. The results suggest that it may introduce noise that alters the dog’s performance, since French dogs had lower success rates than Lebanese dogs. In future studies, targets and controls should at least be paired for sex and age. Another even more important limit is the fact that COVID-19 negative donors were partly people hospitalized for another reason than COVID-19. Meaning that they probably are on medication, and on medication that differs from the one taken by COVID-19 positive patients. How it may impair dogs’ performance or bias there responses towards the COVID-19 positive sample is currently difficult to predict in the absence of further information regarding patients’ treatment.

2) Mocks. The authors now explain that there were 3 or 4 cones in the lineups, including one COVID-19 positive sample and zero, one or two mocks. First, I assume this was zero or one for the 3-cones lineups, and zero to two mocks for the 4-cones lineups, i.e. there was always at least one COVID-19 negative sample in the lineup. Please confirm and write it explicitly in the text. Second, what was the rationale for using mocks? This is potentially a problem, since the test is much easier for the dog when they have 2 mocks plus 2 body odors (i.e., one positive and only one negative sample): they actually are discriminating between 2 and not 4 samples (the level of successfully answering by chance is much higher), since they were trained before to discriminate sweat samples from mocks (step 3 of the training).

3) Where was the dogs’ testing performed? Why did one dog have only 3 cones, and not 4 as the others?

4) Repetition of sample presentations during testing may also be an issue, as raised by the other reviewers. The authors response to this limit was not convincing: rather, they could test whether the performances increased on the second or third repetition compared to the first presentation. In fact, the results section is very short: the authors could very well test some of the possible bias arising from methodological flaws and present them here. For example, in addition to testing the effect of sample repetition, I suggest that they i) test whether lineups including only COVID-19 positive and healthy COVID-19 negative (no sick COVID-19 negative) elicit higher/lower performance, ot ii) test whether one sampling method provided better performances than the other. On the latter point, the fact that “two sampling supports were used because at the time to the beginning of the proof-of-concept study, we did not know which one of the two sampling supports was the most effective” should appear in the manuscript, otherwise no one will understand why you did so. This study being a proof-of-concept study, this is fine to mention that. And it would be nice to have some info regarding effectiveness of both methods.

5) Last paragraph of introduction: you mention the recommendations on how to proceed to rigorously test your research question but we expect to read how you study is positioned within these recommendations. This is currently not the case.

6) There are erroneous references to the bibliography. First, page 4 “sweat does not appear to be a route of excretion for SARS-CoV-2 virus [44, 45]”. In these papers, sweat is not mentioned. Instead, it appears that in the absence of scientific study on that topic, we don’t know whether sweat is able to convey the virus or not, see the following paper: Propper, R.E., 2020. Is sweat a possible route of transmission of SARS-CoV-2? Second, page 6: “SARS-CoV-2 virus does not survive more than a few hours on cotton and disposable gauze [52]. A more recent study concludes that absorbent materials like cotton and gauze are safer than no-absorptive materials for protection from SARS-CoV-2 infection [53].” References 52 and 53 are about SARS-CoV and not SARS-CoV-2, and about the comparison between virus survival on cotton gown versus disposable (polypropylene) gown. Reformulation is needed.

Overall, to make this study publishable, I would recommend to formulate the discussion and abstract even more cautiously. The conclusions are still too affirmative according to the numerous limitations. It should be explicitly stated that more research is need in the future, addressing a number of limitations present in this study – and then all limitations should be carefully listed. Also, the manuscript has clearly not been proofread by an English native speaker, as many language and wording imperfections remain.

Reviewer #2: Review of the Revision R1

The Authors have thoroughly addressed my remarks and comments, both in their responses to my review and in the revision. The plenty of changes to the manuscript makes the tracking of them hardly possible, thus the revision could be considered almost as a new manuscript. Obviously the revision substantially benefits from such considerable changes.

The Authors are aware that in order to avoid confounding factors, control samples must be in all aspects comparable to positive samples, except for the disease status. This was now evident from the lines 131-132, lines 142-145 and 350-365 of the revision.

In lines 371-373 of the revision the Authors properly address the problem of differences in the amount of sweat collected from COVID positive and negative donors, due to possible fever, which is a typical COVID-19 symptom. This issue was my main concern. In the revision the authors inform that the amount of sweat was not recorded and it can not be excluded that it could be a confounding factor. This statement is correct, however, further discussion in lines 374-378 including citation of Concha et al. (2019), is not quite pertinent to the problem I would like to point out. In the study of Concha et al. (2019) on detection threshold of amyl acetate, the dogs had to discriminate between amyl acetate diluted in mineral oil at different concentrations and pure mineral oil as control. There were defined qualitative chemical differences between target odour and control odour up to a dilution of amyl acetate which was for dogs not distinguishable from mineral oil. On the contrary, in the present study it is still not known which volatile organic compounds in the sweat of COVID-positive patiens make that the dogs are able to discriminate the sweat odour of COVID positive from negative humans. Also, the saturation of olfactory receptors by a great amout of the odorant (revision lines 377-378), is not pertinent to the problem of the possible differences in the amount of sweat in the samples. I wanted only to point out that potential differences in the amount of sweat could be a confounding factor and/or could be a cue for dogs. I suggest to delete the passage in revision lines 374-378.

An important information was given only in the response to my review, namely that the samples contained approximately 75 mg sweat and no differences were found between positive and negative samples e.g. due to more sweating in fever. In order to avoid doubts of the readers, this information should be given not only in responses to the review, but first of all in the revision, in the paragraph dedicated to sample collection,.

In the revision the Authors have resigned from the 4 Ajaccio dogs and 21 sweat samples collected in Ajaccio. This was convincingly justified in Authors’ response to the Reviewers #1 and #3. As to the number of dogs used, the Authors in their explanations to the Reviewers #3 inform that now they used six dogs (for the real tests) and they did not mention anymore the number of 14 dogs used for the training. The Authors now explain that „…it added a non relevant information…”. However, in my opinion it is just an importat information, because it shows that not all dogs are able to meet the training or qualification criteria for this task. An information on how many dogs out of all candidates that started the training, were qualified for the test, would be useful for other researchers or dog trainers who will intend to continue or to repeat such experiments.

As to the qualification criterion, the Authors’ information that „…The dog was considered as trained and ready for the testing session by the subjective opinion of the handler, based on the behavior of the dog….” – is acceptable, though for the sake of scientific accuracy in future experiments more precise quantitative criteria would be recommended.

It is not clear to me why the Authors think the calculation of the sensitivity and specificity was not possinle ? (revision lines 276-277).

It depends on how the outcome of a trial in the lineup is assessed. The authors apparently considered their lineup as a simultaneous lineup, where the dog makes a choice out of 3 or 4 samples. This approach seems to be more „convincing” because the theoretical probability of the correct indication by chance is relatively low (1/4 or 1/3). However, the actual probability of correct choices by chance depends on how many samples the dog had really sniffed. The not sniffed samples do not count for calculation of correct hits by chance. Therefore the Authors are not correct, claiming in their explanations that this probability is still 25% if the dog hits correctly on the first cone, without sniffing the remaining cones. Thus, the probabilities given in the last right-hand column of the Table 2 are not constant but variable from trial to trial.

The other approach is a sequential lineup where the dogs are considered to make separate and independent yes/no decisions to each sniffed sample, at the constant 50% probability of correct hits by chance. The sequential lineup is believed to be statistically more sound. If the Authors would consider their lineup as the sequential one, all sniffed but not indicated cones with negative samples are „true negative”, all correct hits on positive samples can be regarded as „true positive”, and the indications labelled in Authors’ protocols as “failures” - are „false positive”. Since in the present protocol „…the dog was previously trained to detect one and only one COVID-19 positive cone, the dog was left free to sniff the cones before marking one cone….” and „….Once a cone was correctly or incorrectly marked, the trial was stopped..(revision lines 258-260) – it means that in the protocol like this, there were no „false negatives” (f.n.=0). Assuming that f.n. =0, the calculated sensitivity would be 100% [Ʃ true positive / Ʃ (true positive + false negative). The specificity for some dogs would be <100% [= Ʃ true negative / Ʃ (false positive + true negative)]. However, from the statement in revision lines 258-260, one can suspect that there was a bias toward reducing „false negative” responses (a miss to indicate the target sample), since the handler knew, and the dogs were trained, that in each trial one of the samples must be the target to be indicated. A problem would be when it comes to testing samples of unknown health status in real scenario. The way to avoid this problem would be either to introduce from time to time a „zero” or „blank” trial (with all negative samples in the lineup), or to set a limit for how many times the dog is allowed to sniff positive samples without indicating, before it eventually hit them. For example if the dogs has sniffed the positive sample 2 or 3 times without indicating, the outcome can be classified as the „false negative”.

Since the aim of the study was only to confirm that dogs are able to discriminate COVID positive and negative sweat samples, the term „Success rate” used by the authors, is acceptable, but for the sake of scientific accurracy in future studies the Authors should be able to calculate the sensitivity and specificity.

It is worth underlining that the revision is better structured than the first submission. The concluding remarks are more cautious and the references are updated.

Minor remarks

The terms „odour detection” and „odour discrimination” are frequently used interchangeably, however, in the context of this work I suggest to use the wording „odour discrimination” rather than „odour detection”. Odour detection refers rather to searching for unknown location of odour source in a field setting. In the odour lineup like this, the dogs are trained to visit the odour locations (cones) which are visible, and their task is to compare and to discriminate between the target and control odours.

In lines 344 and 352 the year of the publication should be completed for more clarity. This is particularly important when two or more papers of the same first author are cited.

Linie 373 „COVID-19 negative” should be inserted for more clarity.

To sum up, only few amendments (minor revision), as listed above, should be made to the text to be fully acceptable for publication in the PLoS ONE.

Reviewer #3: Thank you to the authors for taking into account the previous comments from the three reviewers, providing very detailed responses to the points raised. The revised manuscript is much improved, provides a more balanced appraisal of the study and is much clearer to read. Some of the additional information that has been added has raised further questions and comments, detailed below, but I think the manuscript would be appropriate for publication if these can be addressed.

General comments

- The general use of English language and grammar has greatly improved in this revision. However, there are still a few typographical errors and language issues in places that need addressing with a thorough proof reading.

- Throughout the manuscript (title, abstract, discussion etc.) suggest you clarify that you were testing a discrimination of *symptomatic* SARS-CoV-2 positive vs asymptomatic SARS-CoV-2 negative. This is an important distinction, as symptomatic individuals may produce confounding odours for a wide variety of reasons.

Abstract

- Line 49 “The aim … was to evaluate if a sample of sweat produced by COVID-19 50 positive individuals (SARS-CoV-2 positive) can be detected by trained detection dogs among 51 sweat samples produced by COVID-19 negative individuals.” This wording suggests that dogs were evaluated using just a single (one) SARS-CoV-2 PCR positive sample. The line needs clarifying; I think you mean to say something like: “evaluate if dogs could detect individual samples of sweat produced by COVID-19 50 positive individuals (SARS-CoV-2 PCR positive) presented in a line up amongst sweat samples produced by COVID-19 negative individuals”?

- I think you need to state in the abstract that there are limitations to the study (e.g. processes for collecting positive and negative samples were not completely identical, the trial stage was not fully double blind, some samples were re-used several times during the trial stage) and that a more rigorous validation study is ideally needed to confirm the results.

Introduction

- For balance, I think you should mention that not all studies testing canine detection of disease have shown positive results. There is currently very little evidence relating to canine detection of virus odours, particularly using live subjects (you cite one example using cell cultures).

Materials and Methods

Samples

- Line 157. I think you need to briefly discuss in your discussion section/conclusions the possible limitation that SARS-CoV-2 positive samples were collected using full PPE whereas negatives were not. It could have produced a consistent difference in odour that confounded the results. Were the same brands of gloves used for collecting positive and negative samples? If different brands/types were consistently used for positives and negatives, it’s a major limitation as the dogs may have trained on different glove odours.

- Line 177. Were the two coolers identical (same brand, from the same source etc.?). Worth stating if this was the case. If not, it’s a limitation as dogs may have trained on the odour of two different coolers.

Safety of the dogs regarding SARS-CoV-2 virus infection

- Line 216. Should read “non-absorptive”

Testing protocol

- Line 248. Please clarify the minimum number of SARS-CoV-2 negative samples that were placed out each time. If zero negatives samples were placed out, that would not be valid discrimination data and should be excluded from the analysis/results. If 1, 2 or 3 samples were placed out, please clarify how this was accounted for in the statistical analysis.

Statistical analysis

- Line 275. As per the abstract, this wording suggests that dogs were evaluated using just a single (one) SARS-CoV-2 positive sample. The line needs clarifying.

- As mentioned in the testing protocol comment, some additional detail is needed on how the statistical analysis accounted for the different number of negative samples placed out. If some line-ups included zero negatives, these trials should be excluded from the analysis. It is vital that the analysis is based on the number of negatives placed out, not the number of cones. Clearly, dogs would have a greater chance of detecting the positive samples with lower numbers of negatives present. If not all the cones contained negatives the statistical analysis needs to be revised to account for this.

Results

- Line 292. The ages are the wrong way around?

Table 2, as stated above, the random choice proportion should be based on the number of negatives placed out, not the number of cones (if not all cones (expected the positive) contained SARS-CoV-2 negative samples).

Discussion

- Line 335. This line needs revising to reflect that only positive and negative samples (not “mocks”) should have been included in the statistical analysis.

- line 389 to 396. Recommend you clarify that the lack of double blinding remains a limitation that should ideally be addressed in a future validation study. Previous studies have used various techniques to reward dogs whilst remaining double blind (e.g. use a live mobile phone conversation with an independent referee who has a list of sample “identities” for each line up). The ideal validation study would be to receive samples from an independent source (e.g. from another organisation or even country) which have not been processed by your research team before being presented to the dogs.

- 411 to 416. Recommend key improvements for the validation trial should include double-blinding and samples prepared and delivered by an independent organisation/source.

- Line 412. Should read ‘sensitivity’ not ‘sensibility’.

7. PLOS authors have the option to publish the peer review history of their article (what does this mean?). If published, this will include your full peer review and any attached files.

Reviewer #1: No

Reviewer #2: No

Reviewer #3: No

---

## [Author Response · Author response to Decision Letter 1]

29 Oct 2020

Dear Editor, 

Please find the revised version of our manuscript entitled “Can the detection dog alert on COVID-19 positive persons by sniffing axillary sweat samples? A proof-of-concept study”. It has been modified according to the comments of the three reviewers, and has been revised by a native English speaker.

We do hope that this version adequately answers the points raised by the reviewers.

Furthermore, I would like to modify the order of the co-authors, by placing Loic Desquilbet as the last author because of his significant investment in the writing of the manuscript as well as in the statistical analysis and interpretation of the results. I would like to place Jean-Pierre Tourtier as the co last-author if possible, or as the second author if it is not. 

With my kind regards, 

Prof. Dominique Grandjean

---

## [Decision Letter · Decision Letter 2]

17 Nov 2020

Can the detection dog alert on COVID-19 positive persons by sniffing axillary sweat samples?

A proof-of concept study

PONE-D-20-17256R2

Dear Dr. grandjean,

We’re pleased to inform you that your manuscript has been judged scientifically suitable for publication and will be formally accepted for publication once it meets all outstanding technical requirements.

Kind regards,

Nadine Ravel

Academic Editor

PLOS ONE

Additional Editor Comments (optional):

Reviewers' comments:

Reviewer's Responses to Questions

**Comments to the Author**

1. If the authors have adequately addressed your comments raised in a previous round of review and you feel that this manuscript is now acceptable for publication, you may indicate that here to bypass the “Comments to the Author” section, enter your conflict of interest statement in the “Confidential to Editor” section, and submit your "Accept" recommendation.

Reviewer #1: All comments have been addressed

Reviewer #3: (No Response)

2. Is the manuscript technically sound, and do the data support the conclusions?

Reviewer #1: Yes

Reviewer #3: Yes

3. Has the statistical analysis been performed appropriately and rigorously? 

Reviewer #1: Yes

Reviewer #3: Yes

4. Have the authors made all data underlying the findings in their manuscript fully available?

Reviewer #1: Yes

Reviewer #3: Yes

5. Is the manuscript presented in an intelligible fashion and written in standard English?

Reviewer #1: Yes

Reviewer #3: Yes

6. Review Comments to the Author

Reviewer #1: In this second revision of their manuscript, the authors have answered all my comments in a satisfactory manner. To me, the paper is now acceptable providing some minor changes as detailed below:

1) Ethical statements (page 5) should be formulated as proper sentences (or as such but in a separate section of the manuscript before the references, according to the editor’s preference).

2) Figures 1 and 4 are not required within the manuscript, they may appear either as Appendices at the end of the manuscript or as supplementary materials, again according to the editor’s preference.

3) As early as in the method section, it should be stated explicitly that, within a trial in a line of cones, samples are from the same hospital (are they?) and are made of the same material (gauze swabs or gauze filters). Line 190: it may be useful to explicitly say that polymer tubes were not used in the present study but collected for chemical analysis not presented here.

4) There seem to be a contradiction between Line 193 stating an average quantity of sweat secreted and Line 441 stating that amount of sweat was not recorded. Please address this issue.

Reviewer #3: Thank you to the authors for revising the manuscript and taking into account the comments of the three referees. Overall, I feel the manuscript has been improved again and now provides a clear and balanced discussion of the study and the possible limitations. I have made a few minor comments below, but otherwise would be content for it to be accepted/published – many thanks.

Minor comments:

Introduction

Line 82: The wording “was good compared with standard diagnostic methods” should be clarified (e.g. to “was comparable with” or “was better than” or “was more accurate than” – whichever term explains the comparison more clearly).

Line 126: Suggest change to VOCs “_may_ be detectable by the dogs” (we cannot say for certain that they “are”).

Methods

Line 191: You say two sampling methods were used but three are discussed above (gauze swabs, gauze filters, polymer tubes)?

Discussion

Lines 459-461: “Although the data recorder was not blinded to the sample location, he remained at the back of the room and could not be seen by either the dog handler or the dog during the session. Therefore, he could not have influenced the dog’s behaviour.”

Because the data recorder put out the samples there is always a risk he/she unintentionally influenced the dogs’ choices. There are many theoretical ways this could happen (e.g. they accidentally touched the COVID positive samples more than the negatives so they smell more strongly of the data recorder; they always put the positive samples out first and they had more time to release odour than the negatives etc. etc.). Fundamentally, if the team knows the sample locations there always the risk they may influence the dogs in some way they do not realise and cannot possibly predict – this is the basis of the Clever Hans Effect. That is why double-blind testing is so important for validation of these studies. I suggest you change the text to clarify that standing at the back of the room reduced the chance of the data recorder directly influencing the dog’s behaviour, but the risk of them having some unintentional influence on the choice of samples cannot be completely ruled out (without the double-blind validation study).

7. PLOS authors have the option to publish the peer review history of their article (what does this mean?). If published, this will include your full peer review and any attached files.

Reviewer #1: No

Reviewer #3: No

---

## [Editor Report · Acceptance letter]

24 Nov 2020

PONE-D-20-17256R2 

Can the detection dog alert on COVID-19 positive persons by sniffing axillary sweat samples? A proof-of-concept study 

Dear Dr. Grandjean:

I'm pleased to inform you that your manuscript has been deemed suitable for publication in PLOS ONE. Congratulations! Your manuscript is now with our production department. 

Kind regards, 

on behalf of

Dr. Nadine Ravel 

Academic Editor

PLOS ONE